# Water Stress Alters Morphophysiological, Grain Quality and Vegetation Indices of Soybean Cultivars

**DOI:** 10.3390/plants11040559

**Published:** 2022-02-21

**Authors:** Cássio Jardim Tavares, Walter Quadros Ribeiro Junior, Maria Lucrecia Gerosa Ramos, Lucas Felisberto Pereira, Raphael Augusto das Chagas Noqueli Casari, André Ferreira Pereira, Carlos Antonio Ferreira de Sousa, Anderson Rodrigo da Silva, Sebastião Pedro da Silva Neto, Liliane Marcia Mertz-Henning

**Affiliations:** 1Federal Institute Goiano, Campus Cristalina, CEP, Cristalina 73850-000, GO, Brazil; cassio.tavares@ifgoiano.edu.br; 2Brazilian Agricultural Research Corporation—(EMBRAPA Cerrados), Planaltina 73310-970, DF, Brazil; andre.pereira@ifb.edu.br (A.F.P.); sebastiao.pedro@embrapa.br (S.P.d.S.N.); 3Faculty of Agronomy and Veterinary Medicine, University of Brasília, Brasília 70910-900, DF, Brazil; 4Federal Institute Goiano, Campus Posse, Posse 73900-000, GO, Brazil; lucas.felisberto@ifgiano.edu.br; 5Institute of Geociences, University of Brasília, Brasília 70910-970, DF, Brazil; casari.raphael@gmail.com; 6Brazilian Agricultural Research Corporation, (EMBRAPA Meio-Norte), Teresina 64008-780, PI, Brazil; carlos.antonio@embrapa.br; 7Federal Institute Goiano, Campus Urutaí, Urutaí 75790-000, GO, Brazil; anderson.silva@ifgoiano.edu.br; 8Brazilian Agricultural Research Corporation, National Center for Soybean Research, (EMBRAPA SOJA), Londrina 86001-970, PR, Brazil; liliane.henning@embrapa.br

**Keywords:** NDVI, photochemical reflectance index, gas exchange, automation

## Abstract

Rainfall is among the climatic factors that most affect production, as in the Brazilian Cerrado. Non-destructive and automated phenotyping methods are fast and efficient for genotype selection. The objective of this work was to evaluate, under field conditions, the morphophysiological changes, yield, and grain quality of soybean (*Glycine max* L. Merrill) under water stress in the Brazilian Cerrado. The plots comprised six soybean cultivars and the subplots of four water regimes, corresponding to 31, 44, 64 and 100% of crop evapotranspiration replacement. The experiments were conducted from May to September 2018 and 2019. An irrigation system with a bar of sprinklers with different flow rates was used. Gas exchange, vegetation indices (measured using a hyperspectral sensor embedded in a drone), yield and grain quality were evaluated. Water stress had different effects on gas exchange, vegetation indices, grain yield and chemical composition among the cultivars. Embrapa cultivar BRS 7280 Roundup ready (RR) and Nidera cultivar NA 5909 RG (glyphosate resistant) are yield stable and have a greater tolerance to drought. BRS 7280RR showed a higher tolerance to drought and higher water use efficiency (WUE) than all other tested cultivars. Vegetation indices, such as the NDVI (Normalized Difference Vegetation Index), correlated with the morphophysiological traits, such as plant height, were the most responsive variables to water stress. The NDVI can be used to predict soybean yield as a tool in a selection program under drought.

## 1. Introduction

Soybean is the main cash crop in the Brazilian Cerrado, with positive economic and social effects for the region. It is considered a significant and cheap source of protein, oil and energy for the world, which is essential when considering the challenge of feeding about nine billion people by 2050 [1,2]. Drought is the major factor that affects crop productivity [3,4]. The absence of cultivars tolerant to drought and dry periods in the Brazilian Cerrado [5] are some of the main causes of yield and grain quality losses in the soybean [6]. Therefore, it is essential to find cultivars tolerant to drought [7]. Platforms for drought phenotyping in the field have been successfully used in the Cerrado [1]. The selection of drought-tolerant cultivars can be made through a combination of variables, namely: water use efficiency [8], productivity components [9] and vegetation and physiological indices [10,11,12], which may be correlated with grain yield and quality [13].

The use of vegetation indices (VI) evaluated by multispectral sensors coupled to drones providing wavelength measurements in the visible (VIS; ~400–700 nm) and near-infrared (NIR; ~700–1200 nm) ranges is a rapid, non-destructive assessment and can be used at any stage of plant development for evaluating abiotic and biotic factors [14] and may show a high correlation with yield [15]. These indices are a better option for selecting genotypes compared to invasive, destructive tools in plant–environment interaction studies, as they are related to plant structure, pigments and photosynthetic efficiency and can provide valuable information on the effects of water stress on plant physiology [16].

The normalized difference vegetation index (NDVI) is related to structural traits [15]. However, multiple indices have shown predictive power [17], including VIs associated with structural plant traits (e.g., optimized soil-adjusted vegetation index (OSAVI)); pigment changes (e.g., the absorbance of chlorophyll converted to reflectance index (TCARI) and the ratio between the TCARI and OSAVI, which reduces the effect due to changes in leaf area and soil reflectance). In addition, photochemical activity (PRI) responds to structural changes, pigment content, soil exposure, illumination effects and plant canopy angle [18]. Vegetation indices can be used to select drought-tolerant soybean cultivars [19], while [20] obtained strong relationships between vegetation indices and plant physiological parameters. These indices have been used to select drought-tolerant soybean cultivars and other crops at the leaf level and through UAV (unmanned aerial vehicle) platforms [21,22,23]. In addition, UAV-coupled thermal sensors can measure other important features related to canopy temperature, with small differences in leaf temperature being associated with water stress [21,24].

Morphophysiological traits also are indicators of plant performance under water stress, with decreases in photosynthesis rate, plant height, number of leaves, pods and shoot dry weight [1,25]. In addition, grain quality is altered, increasing protein and decreasing oil content [1,26].

Therefore, it is important to obtain genotypes adapted to stressful conditions, with the ability to grow in periods of drought without significant damage to productivity [10]. Thus, the objective of this work was to validate high-throughput sensors and morphophysiological measurements as tools for the selection of soybean cultivars for yield and grain quality under water stress. This work hypothesizes that reducing water availability reduces grain yield and quality, and non-destructive variables are important for soybean genotype selection under water stress.

## 2. Results and Discussion

### 2.1. Variable Contributions in the Multivariate Response

The mean values of the vegetation indices, morphophysiological evaluations, grain quality and yield in 2018 and 2019 are presented in Appendix A. In the joint multivariate analysis of variance based on singular value decomposition (SVD), differences were found between cultivars (*p* < 0.01), water regime (*p* < 0.01), year of cultivation (*p* < 0.01) and for the interaction cultivars × water regime (*p* < 0.01). The cultivar × water regime × cropping year interaction was not significant (*p* = 0.09). The interaction for most traits showed that cultivars respond differently to water availability. Considering the aim of the present study, this result is an opportunity to validate the selection methodology and identify genotypes adapted to growth under water stress.

In the biplot representation of the decomposition into singular values (Figure 1), the main coordinate 1 (latent variable) retained 56% of the multivariate variation in water regimes and cultivars, while coordinate 2 retained 9.9% of the variation. The biplot provides a useful tool of data analysis and allows the visual appraisal of the structure of large data matrices. It is especially revealed in a principal component analysis that the biplot can show inter-unit distances and indicates the clustering of units, as well as displays variances and correlations of the variables [27]. According to [28], at least 60% of the total variance must be explained by the first two principal components because the information is more concentrated, making interpretation easier.

Water regimes had a greater influence on the multivariate response than cultivars, with a strong contrast between cultivars in WR1 (31% CET replacement) compared to cultivars in WR4 (100% CET replacement). In WR3 and WR2 (64 and 44% CET replacement, respectively), soybean genotypes generally had intermediate values in all studied variables (Figure 1).

Variables whose vectors are in the same direction or where the cosine of the angle between them is close to one have a strong positive correlation (Figure 1). For example, the variables WUE and protein content are correlated positively (Figure 1). Similarly, the variables on the negative number of the graph (left side), i.e., vegetation indices (NDVI, GNDVI, DVI, NDRE, SAVI, PRI, OSAVI and TCARI) and gas exchange (*A*, g*s*, *E*) are correlated positively (Figure 1) and showed similar contributions to the variability of treatments (Coord1 axis), with coefficients close to 1.3 (Figure 1). In contrast, the WUE and net assimilation of CO_2_ (*A*) showed a strong negative correlation (−0.75, *p* < 0.01) (Figure 1 and Figure 2). There was no correlation (or a weak correlation) between plant height and grain yield, as they present an angle close to 90 degrees. The length of the arrows indicates the most important variables in the study to differentiate treatments.

The variables related to the vegetation indices, gas exchange and WUE are important in distinguishing between treatments, as they have higher weights (length of arrows). This indicates that these variables are most indicative of crop performance under water stress [23]. On the other hand, stem diameter, oil content, hectoliter weight, germination and grains per pod showed low weight in the distinction between water regimes and cultivars (Figure 1). For example, cultivars in WR4 have higher vegetation indices and photosynthetic activity, resulting in higher grain yields. On the other hand, cultivars in WR1, under severe water stress, have higher indices in variables such as WUE, percentage of protein in grains and TO (TCARI/OSAVI) (Figure 1).

The lowest indices of these cultivars in WR1 were obtained in variables represented by vectors in the opposite direction, such as NDVI, SAVI, DVI, GNDVI, NDRE, TCARI, OSAVI, PRI, *A*, g*s*, *E* and Fv/Fm. TO is a spectral predictor of canopy-level pigment concentrations that is sensitive to chlorophyll fluctuations and resistant to the effects of ground reflectance and non-photosynthetic materials [29].

The results presented in Figure 3 show that the models are suitable for explaining the variation of the main coordinate 1 (latent variable) in all cultivars. The increase in the value index is exponential and negative and differs among soybean cultivars. Overall, increased water availability increased grain yield, photosynthesis activity and vegetation indices and decreased WUE and iWUE (Figure 4, Figure 5 and Figure 6, Appendix A).

An example of the difference in response between cultivars is the discrepancy between the cultivar BRS 7180IPRO in WR1 and the other cultivars, which responded to water availability (Figure 3). This is confirmed in the regression models for Coord1, which is a linear combination of all response variables. This shows that BRS 7180IPRO responded 1.5 times faster than the BRS 7280RR variety and 1.34 times faster than BRS 5980IPRO. Therefore, changes in water availability will bring important changes in all these variables represented by the latent variable.

### 2.2. Grain Yield and Net CO_2_ Assimilation

In general, cultivars showed exponential responses to grain yield (Figure 4a) and net CO_2_ assimilation (Figure 4b) as a function of water regime, i.e., increasing water availability increases plant photosynthetic activity and promotes better grain formation and grain filling [30]. The BRS7180IPRO genotype had response in grain yield with the addition of water from WR1 to WR2 but from this water level (WR2), BRS7180IPRO and BRS59801IPRO did not respond to irrigation and could be planted in regions with lower rainfall (Figure 4a).

Regardless of water availability, cultivars BRS 7280RR and NA 5909RG had higher grain yields and better or similar results for WUE, iWUE, photosynthesis (*A*) and NDVI within each water regime (Appendix A). Thus, these cultivars have high yield performance under higher and lower water availability and can be recommended for cultivation in the Cerrado region.

On the other hand, BRS 5980IPRO and BRS 7180IPRO showed a 14% decrease in NDVI and WUE, while M6410IPRO and BRS 7380RR showed an intermediate response, close to 6% (Appendix A). The ability to maintain grain yield and quality under water deficit conditions may be related to the strong association between photosynthate assimilation and better carbohydrate mobilization by drought-tolerant genotypes [31]. Furthermore, water deficits lead to changes in plant metabolism that affect productivity, depending on the degree of stress, genotypes and the influence of environmental conditions [32].

In general, the photosynthetic rate gradually decreased among cultivars with a reduction in water availability (Appendix A, Figure 4b). For example, when comparing high (WR4) and low water availability (WR1), there was a reduction in the photosynthetic rate of 70% among the soybean cultivars. Similar results were obtained by [33], where the reduction in water availability caused a decrease in the photosynthetic rate, stomatal conductance, and transpiration, as these processes are highly correlated (Figure 2).

The photosynthetic response of the cultivars in relation to water availability is similar, as the regression coefficients (b) show values close to the mean (0.11) (Figure 4b). In WR1, cultivars showed similar values for net CO_2_ assimilation (Figure 4b). However, with the reduction of water availability, the responses of the cultivars differed, and BRS7280RR and BRS7180PRO showed greater efficiencies in CO_2_ assimilation in all water regimes.

Cell opacity and the proportion of open stomata were affected in plants under water deficit, leading to a reduction in the transpiration and CO_2_ assimilation rates [34]. The transpiration rate of the cultivars was affected by water deficit, as it is regulated by the opening and closing of stomata. Thus, when water availability in the soil decreases, the transpiration values decrease, due to stomata closure. G*s* is considered to be a sensitive indicator of water deficit in plants, so this variable can be used to select genotypes in environments with limited water availability. In addition, it is considered one of the most important factors limiting photosynthesis [35].

When stomata are opened, they allow the assimilation of CO_2_ and the loss of H_2_O. When stomata are closed, CO_2_ entry into the RuBisCO carboxylation sites in chloroplasts is reduced, and H_2_O is conserved, reducing the risk of dehydration but resulting in a loss of the net photosynthetic rate [36]. In (Refs. [37,38]), the authors reported a reduction in the relative leaf water content, stomatal conductance, substomatal carbon dioxide concentration, transpiration rate and photosynthetic rate of soybean grown under water stress.

### 2.3. Water Use Efficiency

Water use efficiency refers to the grain yield obtained per unit of water used [39]. This is a basic physiological/agricultural parameter that indicates the ability of cultivars to conserve water under drought, as it combines drought tolerance and high yield potential, thus having a practical benefit through the efficient use of available water [33]. The cultivars generally showed the same response pattern, and the highest WUE occurred between WR1 and WR2 regardless of the cultivar (Figure 5a, Appendix A). However, there was lower WUE with WR3 and WR4, possibly due to water loss through percolation [8]. According to [40,41], plants under water restriction conditions increased WUE because only a partial reduction of the stomatal aperture restricts transpiration more than CO_2_ influx, which increases WUE. The BRS5980IPRO genotype had the lowest WUE, although not the lowest iWUE, which is more a physiological assessment than an agronomic one. This physiological evaluation was higher in the BRS7280RR genotype than in the others, showing good photosynthetic capacity. Considering the data obtained by iWUE (*A*/g*s*, intrinsic water use efficiency), a higher efficiency was obtained at the leaf level in the treatments with lower applied water (Figure 5b). In [41], the authors reported that high-yield soybean varieties under drought cope with the water shortage by enhancing their photoprotective defenses and promoting growth and productivity, and these processes are linked to a higher intrinsic water use efficiency. Photosynthesis and iWUE are traits to be used in genetic improvement strategies [42].

The cultivars BRS5980IPRO and BRS 7180IPRO showed a maximum productivity of about 3300/3400 kg ha^−1^ in winter cropping and reached this productivity in response to water availability, with about 45% of plant evapotranspiration (Figure 4a). The response for productivity was 2.6, 3, 3.9, 4.1 and 3.6 times faster than the BRS5980IPRO, BRS7280RR, BRS7380RR, M6410IPRO and NA5909RG cultivars, respectively. On the other hand, the BRS7280RR genotype had a higher productivity (Figure 4a) and responded to total water supply with high WUE and iWUE (Figure 5a,b).

### 2.4. Normalized Difference Vegetation Index (NDVI)

The soybean cultivars responded logarithmically to the NDVI variable, i.e., the higher the water availability, the more plants vegetated and increased leaf area (Figure 6). However, the rate at which this response occurred varied among the cultivars.

As with the gas exchange and grain yield, the BRS7180IPRO genotype responded to variations in water availability. An increase of 90% in NDVI was seen when comparing cultivars under the highest and lowest water regimes (Appendix A, Figure 6), showing that water availability affects the vegetation indices of the crop. According to [9,19], plants under stress show changes in spectral responses, and, consequently, leaf area reduces, with an increase in leaf senescence and changes in leaf insertion angle, distribution and spacing. In addition, there are reductions in chlorophyll concentration and photosynthetic activity and a disruption of the internal structures of the leaf, which promotes changes in the vegetation indices. The genotypes with the lowest NDVI at all water levels were NA5909GR and BRS5980IPRO; they showed lower photosynthetic capacity.

The vegetation indices NDVI, TCARI, OSAVI, PRI and GNDVI are efficient to detect maize and soybean plants under water stress conditions [11,16]. Our results showed that, despite being evaluated in only one phenological phase (R5.1), the non-invasive physiological data *A*, g*s* and Fv/Fm showed a high correlation with productivity (Figure 2) but cannot be conducted on a high-throughput scale. In contrast, the vegetation indices GNDI, NDVI, GRVI, OSAVI, NDGI, SAVI and PRI are both high-throughput and non-invasive. These indices showed the highest correlations with yield and can be a useful tool in breeding programs, especially the first two indices (GNDVI and NDVI). The vegetation indices had, in general, a high positive correlation among them (about 0.9) (Figure 2), and one of them could be chosen for the selection of genotypes, except PRI, which presented a lower correlation (approximately 0.5) with the other indices. Among the morphological data, NP and MTG had higher positive correlations with productivity but are time-consuming and are, therefore, unsuitable for large-scale use. WUE and iWUE had negative correlations with yield (*p* < 0.05, about −0.6) (Figure 2) and can only be used for specific purposes not linked to yield.

These results were also reported by [10], where gas exchange measurements and vegetation indices can differentiate the responses of soybean genotypes in relation to water availability, and, in some cases, the indices are more sensitive than gas exchanges to detect the effects of genotype, especially indices that use bands in the infrared range. In (Ref. [15]), the authors reported that the SAVI and NDVI indices are excellent for predicting soybean yields. Regions with the highest values of these indices can achieve the highest grain yields in field conditions, providing an advantage for using a multispectral sensor coupled to an unmanned aerial vehicle.

### 2.5. Protein Content and Oil Content in the Grains

There was a high negative correlation between oil and protein content (*p* < 0.05, −0.70), and lower water availability promoted greater protein accumulation (Figure 2). The highest content of proteins was also observed by [26] in the common bean, with an increase of 20.57% in the lowest water regime (187 mm) compared to the highest water regime (535 mm).

The negative correlation between oil and protein content in grains can also be explained by the competition of synthesis pathways by carbon skeletons, changes in accumulation and the distribution of nutrients in soybean seeds under water stress conditions [13]. The acceleration of maturation and early senescence promotes anticipation of the cycle, a reduction in the photosynthetic period and accumulation of reserves, resulting in the grains of stressed plants not exhibiting the normal pattern of development and chemical composition [43].

This result poses a challenge to geneticists striving to increase oil and protein content, which is desirable for processing high-value soybean products. For example, in (Ref. [13]) the authors found a decrease in protein content, palmitic and linoleic acids, sucrose, raffinose, stachyose, N, P, K and Ca in soybean, while the content of oleic, stearic, oleic, and linolenic acids, Fe, Mg, Zn, Cu and B increased under low soil moisture conditions.

## 3. Materials and Methods

### 3.1. Experimental Design and Conducting the Experiment

The experiment was conducted in Planaltina, DF, Brazil (15°35′30″ S, 47°42′30″ W, altitude of 1006 m). The climate in the region is Aw (Koeppen–Geiger), tropical, with rainfall concentrated in the summer (October to April) and a pronounced dry period during the winter (May to September), with an average annual rainfall of 1200 to 1500 mm. Climatological data were collected in 2018 and 2019 at a meteorological station near the experiment (Figure 7).

The experiment was carried out between May and September, which coincides with the dry season in the region, which allows for controlling the water supply to the plants. The soil is classified as an Oxisol with a clay texture [44], and the soil analysis carried out before conducting the experiment showed the following physicochemical properties at a depth of 0 to 20 cm: pH (CaCl_2_) of 5.7; 11 mg dm^−3^ P; 186 mg dm^−3^ K; 5.77 cmol_c_ dm^−3^ Ca; 1.83 cmol_c_ dm^−3^ Mg; 0.02 cmol_c_ dm^−3^ Al; 15.7 mg dm^−3^ N-NO^−^_3_; 2.6% organic matter and granulometry of 46, 10 and 44% of clay, silt and sand, respectively.

The soil water retention curve, fitted according to model by [45], had the following values: residual water content (θs) 0.0839 cm^3^ cm^−3^, saturated water content (θs) 0.5500 cm^3^ cm^−3^ and parameters α (1.892 kPa^−1^) and *n* (1.2390). The field capacity moisture was 0.3423 cm^3^ cm^−3^. The parameter α is associated with the inverse of the value of matric potential at which air enters the largest pore of the soil, therefore representing a unit of pressure. The parameter *n* is an index of the pore size distribution, therefore related to the slope of the soil–water characteristic curve.

The experimental design was randomized blocks in a split-plot scheme with three replications. Plots consisted of soybean cultivars (NA 5909RG; M 6410IPRO; BRS 5980IPRO; BRS 7180IPRO; BRS 7280RR and BRS 7380RR) and subplots corresponded to four water regimes (WR). The cultivars BRS 5980IPRO (roundup ready and intacta technology), BRS 7180IPRO, BRS 7280RR and BRS 7380RR (roundup ready technology) were developed by Embrapa’s genetic improvement program. They were selected because of their high yield, production stability and broad adaptability to the various grain-producing regions of the Brazilian Cerrado. In addition, the varieties Nidera cultivar (NA) 5909RG (glyphosate resistant technology) and Monsoy cultivar (M) 6410IPRO are also recommended for the region and were used in this work as control genotypes.

In 2018, 160.1 mm, 274.68 mm, 420.55 mm and 634.35 mm were applied during the crop cycle, corresponding to WR1, WR2, WR3 and WR4, respectively. In 2019, 164.79 mm, 237.45 mm, 343.15 mm and 531.43 mm were applied, corresponding to WR1, WR2, WR3 and WR4, respectively. In 2018 and 2019, the irrigation regimes used corresponded to 31%, 44%, 64% and 100% of crop evapotranspiration (CET) replacement. In both years, rainfall was included in the calculations of applied water.

The history of the previous three years of cultivation in the experimental area was soybean under different water regimes in winter and fallow in summer. The area was desiccated 20 days before sowing with glyphosate at a dose of 1440 g.e.a ha^−1^. The soybean seeds were previously inoculated with *Bradyrhizobium japonicum* (strain SEMIA 5080) at a dose of 100 mL per 50 kg of seed. The seeds were sown mechanically on 2 June 2018 and 23 May 2019 under no-tillage systems, with 16 seeds per meter. For basic fertilization, 300 kg ha^−1^ of fertilizer with the formula 04-30-16 (N, P_2_O_5_ and K_2_O) was used.

Phytosanitary treatments were performed for cucurbit beetle (*Diabrotica speciosa*) control; the insecticide thiamethoxam + lambda-cyhalothrin was applied at a dosage of 14.1 g + 10.6 g ha^−1^ on the 10th and 20th day after soybean emergence (DAE) in 2018 and on the 12th and 24th DAE in 2019. In addition, glyphosate was applied at a dose of 720 g.e.a. ha^−1^ for weed control at 18 DAE in both years.

In both experiments, the same water depth was applied during the first 35 DAE (days after emergence), up to the vegetative stage V3, with an average of 140 mm of water applied to obtain a homogeneous plant stand. After this period, the “line source” method was applied [46], modified by the introduction of an irrigation bar [47]. Sprinklers with a decreasing flow from the center to the edge of the bar were used to create a water deficit gradient. The water regime (WR) was achieved by an irrigation bar (IrrigaBrasil model 36/42) with a width of 20 m on each side, connected to a self-propelled TurboMaq 75/GB whose speed was adjustable according to the depth of water to be applied.

Each experimental unit consisted of a genotype formed by 36 cultivation lines 5.0 m long with 0.5 m spacing. Each water regime (WR) was an experimental subunit with a length of 5.0 m, formed by eight lines spaced 0.50 m apart, with the usable area formed by the two central lines, excluding the edges and 0.5 m from each end, that is, 4 m^2^ (Figure 8).

Irrigation at the highest level was carried out as described in the Cerrado Irrigation Monitoring Program [48] to replace evapotranspiration, using agrometeorological indicators of the region and soil type and crop emergence date. The program estimated the reference evapotranspiration based on the equation proposed by Penman and Monteith. The irrigation was carried out approximately every five days, depending on the climatic conditions and the phenological phase of the plants. Two rows of collectors were placed parallel to the irrigation pipe to measure the water applied during each irrigation. Soil water content in the higher and lower water regimes (WR1 and WR4) (Figure 8) just before irrigation and measured at flowering were between 15 (−1500 Kpa) and 24% (−50 Kpa).

### 3.2. Gas Exchange and Fluorescence Analysis

At 70 DAE, the net assimilation of CO_2_ (*A*, µmol CO_2_ m^−2^ s^−1^), stomatal conductance (g*s*, mol H_2_O m^−2^ s^−1^) and transpiration rate (*E*, mmol H_2_O m^−2^ s^−1^) were evaluated in the phenological phase R5.1. This evaluation was performed from 8:30 a.m. to 12:30 p.m. at an irradiance of 1200 μmol photons m^−2^ s^−1^ and an external CO_2_ concentration (C_a_) of 400 μmol mol^−1^ in the air using an IRGA (infra-red gas analyzer) and a portable open-flow gas exchange device (LI-6400xt LI-COR Inc., Lincoln, NE, USA). The chlorophyll fluorescence variable and maximum quantum yield of photosystem II (Fv/Fm) were measured using a modulated portable fluorometer coupled to the IRGA. Evaluations were conducted on dark-adapted leaves for at least 3 h and were performed after 10:30 p.m. The reaction centers were fully opened (all oxidized primary electron acceptors) with minimum heat loss. Under this condition, it was possible to estimate the initial fluorescence (F0), maximum fluorescence (Fm), and maximum quantum yield of photosystem II [Fv/Fm = (F0 − Fm)/Fm)] [49].

Three evaluations were made in each subplot to quantify gas exchange. Evaluations were performed on the youngest fully expanded leaves, with light adapted. Measurements were made under controlled CO_2_ concentration, temperature and H_2_O vapor from the study area, with the reference air homogenized in a 20L container before reaching the leaf chamber. The measurements were performed after the coefficient that combines the variations of carbon dioxide (ΔCO_2_), water (ΔH_2_O) and air flow (Δμe) was below 1%. The relative humidity was between 65 and 70%, the temperature was between 20 and 25 °C (night/day), the irradiance was 1200 μmol photons m^−2^ s^−1^ and the external concentration of CO_2_ was 400 μmol mol^−1^. The intrinsic water-use efficiency (iWUE) was calculated by the ratio between the net assimilation of CO_2_ and the stomatal conductance (*A*/g*s*).

### 3.3. Vegetation Index

Vegetation indices (VI) were determined in phase R5.1 using a Micasense RedEdge multispectral camera model coupled to a rotary-wing unmanned aerial vehicle (UAV). This camera captures images in five different spectral bands, namely blue (range: 465–485 nm; width: 20 nm), green (range: 550–570 nm; width: 20 nm), red (range: 663–673nm; width: 10 nm), red edge (range: 712–722 nm; width: 10 nm) and near-infrared (NIR) (range: 820–860 nm; width: 40 nm), with an optical resolution of 1280 × 960 pixels and images recorded in RAW12 bits [50]. The flight took place at an altitude of 45 m at 10:00 a.m. on the same day as the gas exchange measurements. Reflectance maps were computed from mosaic generation in Pix4D Mapper software (v5.4.6, Pix4D, Lausanne, Switzerland) based on images from the calibration panel (MicaSense, model RP04 CRP) before and after the flight, in addition to the radiation detection at the time of each image. Subsequently, the images were processed using the raster package in R software, and the vegetation indices were extracted (Table 1).

### 3.4. Grain Yield and Quality Components

Grain yield (GY) was evaluated in each experimental subplot. Ten representative plants were randomly harvested in the crop area to evaluate the height of insertion of the first pod (A1), plant height (PH), stem diameter (SD) and the total number of pods per plant (PP). From these plants, 200 pods were randomly collected, and the number of grains per pod (GP) and mass of thousand grains (MTG) were determined. The grain yield and mass of thousand grains were expressed with grain water content standardized to 13%. The water use efficiency (WUE) was calculated using the relationship between grain yield and crop water applied [39]. Additionally, the intrinsic water efficiency (iWUE) was calculated by the relation *A*/g*s*, measuring gas exchange using IRGA.

The grain quality (protein and oil content) and hectoliter weight (HW) were analyzed. The protein and oil content (%) were determined in whole grains and without impurities, according to [60]. The analyses were performed in the Chemical Analysis Laboratory at EMBRAPA Soybean, by Fourier transform near-infrared spectroscopy (FT-NIR, model Antaris II, ThermoFisher Scientific, Waltham, MA, USA), using 30 g samples of grains and using an integrating sphere with readings ranging from 1100 to 2500 nm. The mathematical models developed by EMBRAPA Soja in 2011–2012 were used to predict the protein content, including 180 standards, correlation coefficient (r) = 0.97 and root mean square error of calibration (RMSEC) = 0.64. For the oil content: 170 standards, r = 0.98 and RMSEC = 0.45.

### 3.5. Statistical Analysis

The data were subjected to joint multivariate analysis of variance by harvest based on singular value decomposition (SVD). The residuals were tested for multivariate normality using the generalized Shapiro–Wilk test [61] and for homogeneity of covariance matrices using the Box-M test [62]. In two years, treatment (combinations of cultivar and water regime levels) means were presented graphically in a biplot [27], and this allows visualization of the relationship among genotypes and treatments. A Pearson correlation analysis (*t*-test, *p* < 0.05) was performed with the residuals. Regression models were fitted to unravel the effect of water regime on the response variables, and mean values of the variables were presented in the table (Appendix A). The statistical analyses were carried out using the R v3.6.1 software.

## 4. Conclusions

As water negatively affected various parameters differently between soybean cultivars, there is an opportunity to select genotypes for drought tolerance. As high-throughput parameters could predict soybean yield under water stress, they would be useful selection tools in plant breeding programs. The BRS7280RR genotype showed better drought tolerance and water use efficiency than the other genotypes and are able to be used in irrigated and stressed conditions. With the advent of climate change affecting the Cerrado, this result could be useful in a likely increase in drought events in the region. Overall, the study highlights the potential impact of lower water availability on important traits to farmers and consumers and highlights cultivars less affected by this condition.

## Figures and Tables

**Figure 1 plants-11-00559-f001:**
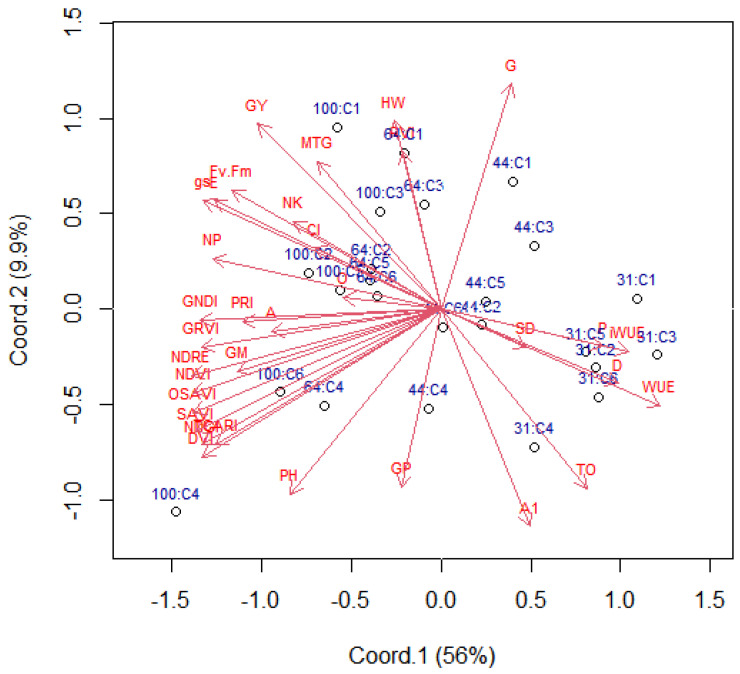
Biplot for mean values of the water regimes (31, 44, 64 and 100% CET replacement, represented by WR1, WR2, WR3 and WR4, respectively) and soybean cultivars (C1, 5909RG; C2, M 6410IPRO; C3, BRS 5980IPRO; C4, BRS 7180IPRO; C5, BRS 7280RR and C6, BRS 7380RR) from the variables: NDVI, normalized difference vegetation index; SAVI, soil-adjusted vegetation index; DVI, difference vegetation index; GNDVI, green normalized difference vegetation index; NDRE, red edge normalized difference; TCARI, the transformed chlorophyll absorption and reflectance index; OSAVI, optimized soil-adjusted vegetation index; TO, TCARI/OSAVI ratio; PRI, photochemical reflectance index; A, net assimilation of CO_2_; g*s*, stomatal conductance; E, transpiration; Fv/Fm, photosystem II maximum quantum yield; CI, internal carbon concentration; A1, height of first pod set (cm); PH, plant height (cm); SD, stem diameter (mm); NK, number of knots; PP, number of pods; GP, grains per pod; GM, grain moisture; HW, hectoliter weight; MTG, mass of thousand grains; GY, grain yield; G, germination; WUE, water use efficiency; O, oil; P, protein; D, hard seeds; iWUE, intrinsic water use efficiency (*A*/g*s*, net assimilation of CO_2_/stomatal conductance).

**Figure 2 plants-11-00559-f002:**
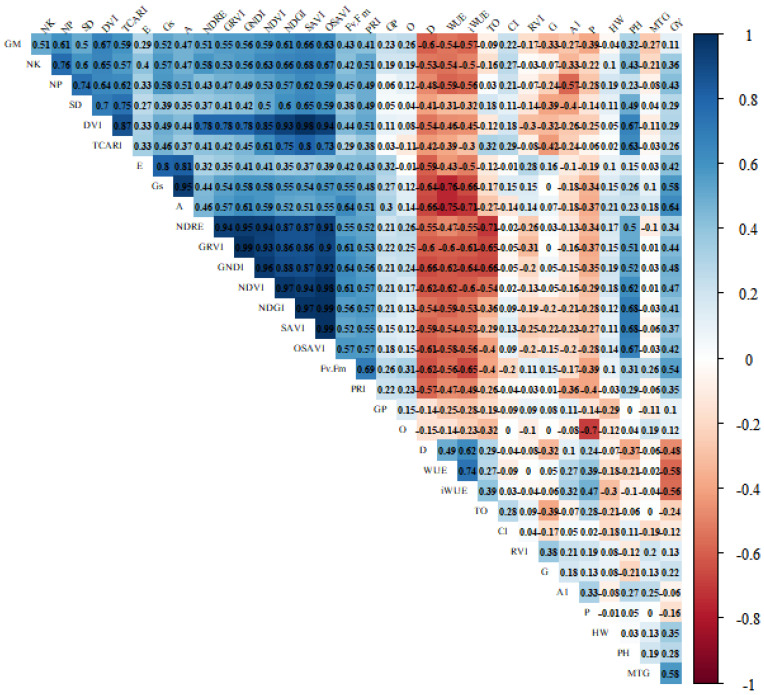
Pearson correlogram between the yield components, vegetation indices, gas exchange and grain quality as a function of the water regimes (31, 44, 64 and 100% CET replacement, represented by WR1, WR2, WR3 and WR4, respectively) and soybean cultivars (NA 5909RG, M 6410IPRO, BRS 5980IPRO, BRS 7180IPRO, BRS 7280RR and BRS 7380RR). GM (grain moisture), NK (number of knots), NP (number of pods), SD (stem diameter), iWUE (intrinsic water use efficiency (*A*/g*s* net assimilation of CO_2_/stomatal conductance)), DVI (difference vegetation index), TCARI (the transformed chlorophyll absorption and reflectance index), *E* (transpiration), g*s* (stomatal conductance), *A* (net assimilation of CO_2_), NDRE (red edge normalized difference), GRVI (green red vegetation index), GNDVI (green normalized difference vegetation index), NDVI (normalized difference vegetation index), SAVI (soil-adjusted vegetation index), OSAVI (optimized soil-adjusted vegetation index), Fv/Fm (maximum quantum yield of photosystem II), PRI (photochemical reflectance index), GP (grains per pod), O (oil content), WUE (water use efficiency), D (hard seeds), TO (TCARI/OSAVI ratio), CI (internal carbon concentration), RVI (ratio vegetation index), G (germination percentage), A1 (height of first pod set), P (protein content), HW (hectoliter weight), PH (plant height), MTG (mass of thousand grains), GY (grain yield).

**Figure 3 plants-11-00559-f003:**
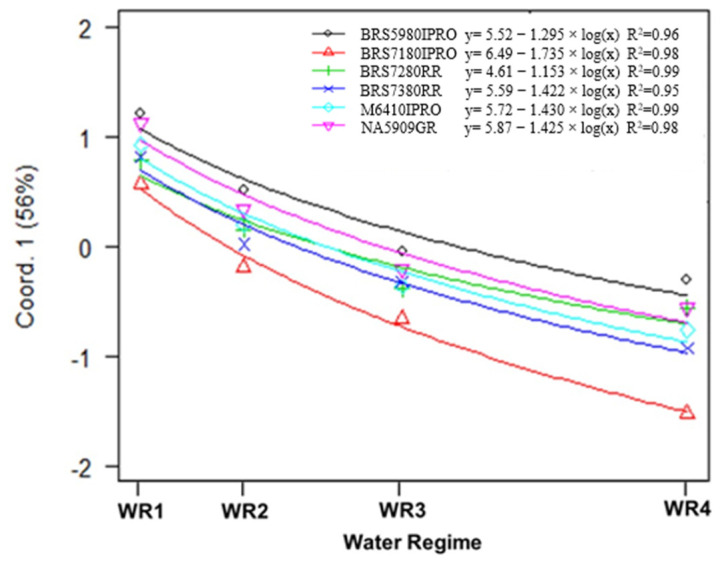
Regression equations and the respective coefficients of determination (R²) of the variable Main Coordinate 1. The latent variable (Coord. 1) as a function of water regime (WR1, WR2, WR3 and WR4, representing 31, 44, 64 and 100% CET replacement, respectively) and soybean cultivars (NA 5909RG, M 6410IPRO, BRS 5980IPRO, BRS 7180IPRO, BRS 7280RR and BRS 7380RR).

**Figure 4 plants-11-00559-f004:**
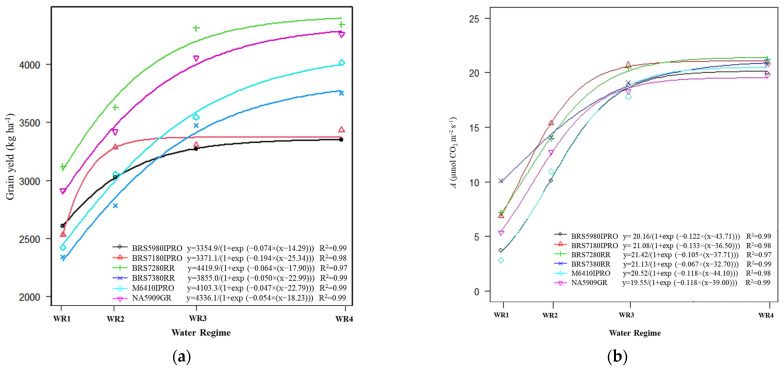
Regression equations and the respective coefficients of determination (R²) of the variables: (**a**) grain yield and (**b**) net CO_2_ assimilation (*A*) as a function of water regime (WR1, WR2, WR3 and WR4, representing 31, 44, 64, and 100% CET replacement, respectively) and soybean cultivars (NA 5909RG, M 6410IPRO, BRS 5980IPRO, BRS 7180IPRO, BRS 7280RR and BRS 7380RR).

**Figure 5 plants-11-00559-f005:**
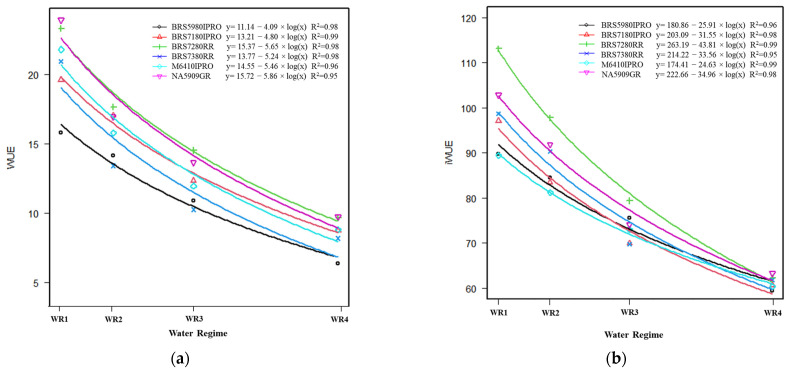
Regression equations and the respective coefficients of determination (R²) of the variables: (**a**) WUE, water use efficiency; (**b**) iWUE, intrinsic water use efficiency (*A*/g*s*, net assimilation of CO_2_/stomatal conductance) as a function of water regime (WR1, WR2, WR3 and WR4, representing 31, 44, 64 and 100% CET replacement, respectively) and soybean cultivars (NA 5909RG, M 6410IPRO, BRS 5980IPRO, BRS 7180IPRO, BRS 7280RR and BRS 7380RR).

**Figure 6 plants-11-00559-f006:**
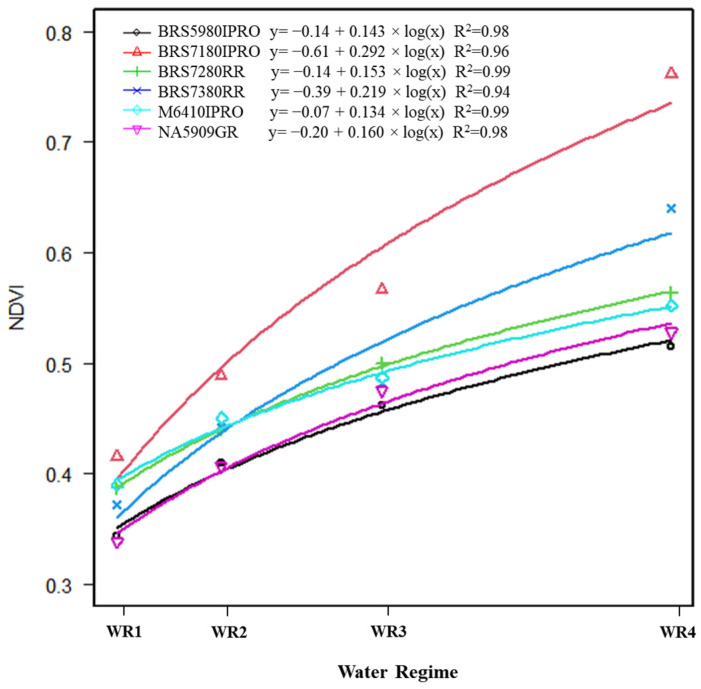
Regression equations and the respective coefficients of determination (R²) of the variable normalized difference vegetation index (NDVI) as a function of water regime (WR1, WR2, WR3 and WR4, representing 31, 44, 64 and 100% CET replacement, respectively) and soybean cultivars (NA 5909RG, M 6410IPRO, BRS 5980IPRO, BRS 7180IPRO, BRS 7280RR and BRS 7380RR).

**Figure 7 plants-11-00559-f007:**
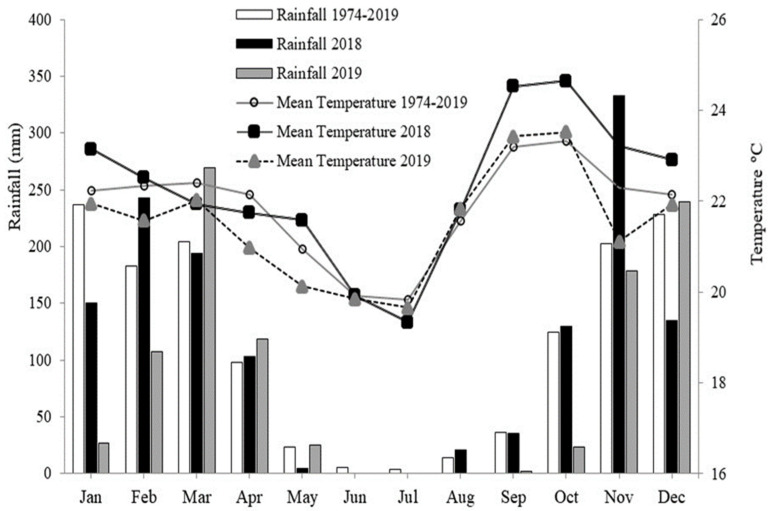
Precipitation and average temperature measured by an automatic weather station near the experiment in 2018 and 2019 and a historical series (1974–2019).

**Figure 8 plants-11-00559-f008:**
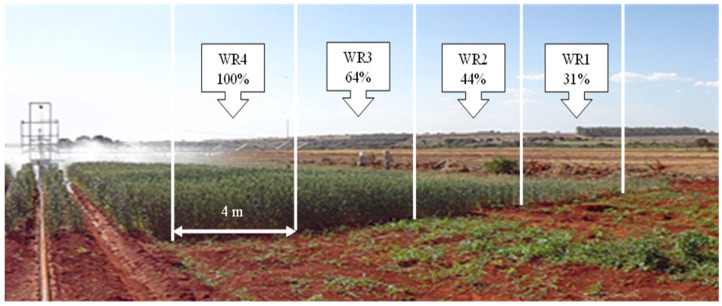
Graphical summary of experimental subunits (WR1, WR2, WR3 and WR4, which represent31, 44, 64 and 100% CET replacement, respectively) and irrigation bar in soybean crop.

**Table 1 plants-11-00559-t001:** Evaluated spectral characteristics and their respective calculation formulas.

Vegetation Index	Formula	Source
Normalized Difference Vegetation Index	$\mathrm{NDVI}=\frac{\left(\mathrm{NIR}-\mathrm{RED}\right)}{\left(\mathrm{NIR}+\mathrm{RED}\right)}$	[51]
Green Normalized Difference Vegetation Index	$\mathrm{GNDVI}=\frac{\left(\mathrm{NIR}-\mathrm{GREEN}\right)}{\left(\mathrm{NIR}+\mathrm{GREEN}\right)}$	[52]
Green-Red Vegetation Index	$\mathrm{GRVI}=\frac{\mathrm{NIR}}{\mathrm{GREEN}}$	[53]
Difference Vegetation Index	DVI=NIR−RED	[54]
Normalized Difference Red Edge	$\mathrm{NDRE}=\frac{\left(\mathrm{NIR}-\mathrm{RedEdge}\right)}{\left(\mathrm{NIR}+\mathrm{RedEdge}\right)}$	[55]
Soil-Adjusted Vegetation Index	$\mathrm{SAVI}=\frac{[\left(1+L\right)\left(\mathrm{NIR}-\mathrm{RED}\right)}{\left(\mathrm{NIR}+\mathrm{RED}+L\right)]}$	[56]
Photochemical l Reflectance Index	$\mathrm{PRI}=\frac{\left(\mathrm{BLUE}-\mathrm{GREEN}\right)}{\left(\mathrm{BLUE}+\mathrm{GREEN}\right)}$	[57]
Optimized Soil-Adjusted Vegetation Index	$\mathrm{OSAVI}=\frac{\left(\mathrm{NIR}-\mathrm{RED}\right)}{\left(\mathrm{NIR}+\mathrm{RED}+0.16\right)}$	[58]
Chlorophyll Absorption and Reflectance Index	TCARI = 3[(RedEdge-Red) − 0.2(RedEdge-Green) (Red Edge/Red)]	[59]
TCARI/OSAVI Ratio	TO = TCARI/OSAVI	[59]

## Data Availability

All soybean cultivars used in this manuscript are released on the market in Brazil registered with the Ministry of Agriculture: https://sistemas.agricultura.gov.br/snpc/cultivarweb/cultivares_registradas.php (accessed on 8 February 2022).

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
