# Peer review of "Water Stress Alters Morphophysiological, Grain Quality and Vegetation Indices of Soybean Cultivars"

_plants, 2022, doi:10.3390/plants11040559_

Round 1

Reviewer 1 Report

Tavares et al. Water stress alters morphophysiological, grain quality and vegetation indices of soybean cultivars

In this paper, the authors studied morphology, photosynthetic gas exchange, yield and grain quality, and vegetation indices under four different water conditions. The results found the influence of water conditions on soybean physiology and production, and suggest the effectiveness of using vegetation indices to indicate crop health and yield. The experiment design is solid and the results will be of interest to agronomists. However, more details in the methodology are needed, and better analyses are needed to show the best vegetation index or combination to predict vegetation health and yield. Some detailed comments are listed below:   

Line 36: please clarify “higher than” what

Line 37: please clarify which vegetative indices were used

Line 49: water use efficiency is a more commonly used term.

Introduction: Why study soybeans? Is soybean an important crop in Cerrado. Please provide more details on the importance of soybeans in the area.

Line 60: NDVI. What structural traits is NDVI related? Please explain in more details.

Line 70-72: This is just the experiment or measurements you did, not the objectives. Please also describe related hypotheses.

Line 75: remove “)” after 2.

Line 82-83: Please explain what analysis this is. PCA or CCA?

Line 247: Or because crops show low leaf-level WUE. You can calculate A/gs to confirm.

Line 278: An analysis is needed to show which index is the best to indicate crop performance under drought.

Line 325-327: please explain all variables including a and n.

Line 330: explain the rationale for choosing these cultivars for this study.

Line 367-373: any measurements of soils water content?

Line 376: maximum quantum yield of photosystem II (Fv/Fm) needs to be measured at predawn when the leaves are fully dark-adapted. Please explain more details on when you did the measurements, and also whether the leaves were dark-adapted or under the light.

Gas exchange measurements: please provide detailed environmental conditions including temperature, relative humidity etc.

Statistical analysis: please also explain the analysis for Figure 1.

Conclusions. Please write the conclusions as a paragraph. More analyses are needed to figure out which one or combination can predict yield best.

Author Response

Manuscript: Water stress alters morphophysiological, grain quality and vegetation indices of soybean cultivars.

Reviewer 1:

1) Line 36 (abstract): Reviewer comment: please clarify “higher than” what “.

Original text: BRS 7280RR showed higher tolerance to drought and WUE”. 

Modified text: Considering yield under drought BRS 7280RR showed higher tolerance to drought and WUE”. 

2) Line 37 (abstract): Reviewer comment: please clarify which vegetative indices were used. 

Original text: Vegetative indices, correlated to the morphophysiological traits, and were the most responsive variables to water stress, which can predict soybean yield and can be used as a tool in selection for this purpose.

Modified text: Line 37 in the new version –

Vegetative indices such as NDVI, correlated to the morphophysiological traits such as plant height, were the most responsive variables to water stress, which can predict soybean yield and can be used as a tool in selection for this purpose.

Author comment: Write all indices would be too long.

3) Introduction: Reviewer comment: Why study soybeans? Is soybean an important crop in Cerrado. Please provide more details on the importance of soybeans in the area.

To be included in the begginning of the introduction:

Lines 45 to 48

Soybean is the main cash crop in the Cerrado with positive economic and social effects for the region and it is considered an important and cheap source of protein, oil and energy for the world, what is important considering a challenge of feed about nine billion people by 2050 [1,2].

4) Line 49: Reviewer comment: water use efficiency is a more commonly used term

Original text: Text: water productivity use efficiency [7], productivity components [8], vegetative and physiological indices [9, 10, 11], which may be correlated with grain yield and quality [12].

Modified text: Line 54 in the new version - The selection of drought-tolerant cultivars can be made through a combination of variables, namely: water use efficiency [8], productivity components [9], vegetative and physiological indices [10, 11, 12], which may be correlated with grain yield and quality [13].

5) Line 70-72: Reviewer comment: This is just the experiment or measurements you did, not the objectives. Please also describe related hypotheses.

Original text: The objective of this work was to evaluate, under field conditions, the morphophysiological changes, yield, and grain quality of soybean under drought in the Brazilian Cerrado.

Modified text: Lines 85 to 90 in the new version -  

This work hypothesizes that reduction in water availability reduces grain yield and quality and non-destructive variables are important for soybean genotype selection under water stress.

The objective of this work was to validate high throughput sensors and morphophysiological measurements as a tool for selection of soybean cultivars for yield and grain quality under water stress.

6) Line 75: Reviewer comment: remove “)” after 2.

Original text: The mean values of vegetation indices, morphophysiological evaluations, grain quality and yield in 2018 and 2019 are presented in Tables 1 and 2).

Modified text: Line 93 to 94 in the new version –

The mean values of vegetation indices, morphophysiological evaluations, grain quality and yield in 2018 and 2019 are presented in Tables S1 and S2.

7) Line 82-83: Reviewer comment: Please explain what analysis this is. PCA or CCA?

Original text: In the biplot representation of the decomposition into singular values (Figure 1), the main coordinate 1 (latent variable) retained 56% of the multivariate variation in water regimes and cultivars, while coordinate 2 retained 9.9% of the variation.

Modified text: Lines 100 t0 103 in the new version –

In the biplot representation of the decomposition into singular values (Figure 1), the main coordinate 1 (latent variable) retained 56% of the multivariate variation in water regimes and cultivars, while coordinate 2 retained 9.9% of the variation. Biplot analysis is a multivariate technique that can be used to analyze interrelationships between a large number of variables and explain these variables in terms of their inherent dimensions and works with one set of variables [27].

Author´s reply: The type of analysis used is called biplot, similar to PCA (Gabriel, 1971). It is a graphical representation of decomposition into singular values ​​(SVD).

Gabriel, K.R. The biplot graphical display of matrices with application to principal component analysis. Biometrika, 1971, 58, 453-467. https://doi.org/10.2307/2334381

8) Line 247: Reviewer comment: Or because crops show low leaf-level WUE. You can calculate A/gs to confirm.

Original text: Water use efficiency refers to the grain yield obtained per unit of water used [29]. This is a basic physiological parameter that indicates the ability of cultivars to conserve water under water scarcity, as it combines drought tolerance and high yield potential, thus being a practical benefit in terms of efficient use of applied water [23]. The cultivars generally showed the same response pattern, and the highest WUE occurred between WR1 and WR2 regardless of the cultivar (Figure 5a, Table 2). However, there was lower WUE in WR3 and WR4, possibly due to water loss through percolation [7]. According to [30], plants under water restriction conditions show an increase in WUE because only a partial reduction of stomatal aperture restricts transpiration more than CO2 influx, which increases WUE.

Modified text: Lines 238]9 to 253 in the new version - 

Water use efficiency refers to the grain yield obtained per unit of water used [39]. This is a basic physiological parameter that indicates the ability of cultivars to conserve water under water scarcity, as it combines drought tolerance and high yield potential, thus being a practical benefit in terms of efficient use of applied water [33]. The cultivars generally showed the same response pattern, and the highest WUE occurred between WR1 and WR2 regardless of the cultivar (Figure 5a, Table S2). However, there was lower WUE in WR3 and WR4, possibly due to water loss through percolation [8]. According to [40, 41], plants under water restriction conditions show an increase in WUE because only a partial reduction of stomatal aperture restricts transpiration more than CO2 influx, which increases WUE. Considering the data obtained by iWUE (A/gs, intrinsic water use efficiency), the higher efficiency at leaf level in treatments with lower applied water was obtained (Figure 5b). [41] report that high-yield soybean varieties under drought cope with the shortage of water by enhancing their photoprotective defenses and invest in growth and productivity, linked to a higher intrinsic water use efficiency. [42] indicate photosynthesis and iWUE as traits to be used in genetic improvement strategies.

 Figure 5. Regression equations and the respective coefficients of determination (R²) of the variables: (a) WUE -water use efficiency, (b)- iWUE - intrinsic water use efficiency (A/gs: net assimilation of CO2/stomatal conductance as a function of water regime (WR1, WR2, WR3 and WR4 - 31, 44, 64, resp. 100% CET replacement) and soybean cultivars (NA 5909RG; M 6410IPRO; BRS 5980IPRO; BRS 7180IPRO; BRS 7280RR; BRS 7380RR).

9) Line 278: Reviewer comment: An analysis is needed to show which index is the best to indicate crop performance under drought.

Original text: The vegetation indices NDVI, TCARI, OSAVI, PRI and GNDVI are efficient to detect maize and soybean plants under water stress conditions [10, 15]. 

Modified text: Lines 287 to 295 in the new version

The vegetation indices NDVI, TCARI, OSAVI, PRI and GNDVI are efficient to detect maize and soybean plants under water stress conditions [11, 16]. Our results showed that the physiological data A and gs had a high correlation with productivity but could not be done on a large scale, while the vegetative indices NDVI and GNDVI which are high throughput were the ones that showed the highest correlation with yield and can be a useful tool in improvement programs. The vegetation indices had, in general, correlation between them, and one of them could be chosen for the selection process. Among the morphological data NP had a greater correlation with productivity but also a time-consuming evaluation (Fig.2).

10) Line 325-327: Reviewer comment: please explain all variables including a and n.

Original text:

Modified text: Lines 334 to 340 in the new version –

The experiment was carried out between May and September, which coincides with the dry season in the region that allows controlling the water supply to the plants. The soil is classified as Oxisol with clay texture [44], and the soil analysis carried out before conducting the experiment showed the following physicochemical properties at a depth of 0 to 20 cm: pH (CaCl2) of 5.7; 11 mg dm-3 P; 186 mg dm-3 K; 5.77 cmolc dm-3 Ca; 1.83 cmolc dm-3 Mg; 0.02 cmolc dm-3 Al; 15,7 mg dm-3 N-NO-3; 2.6% organic matter and granulometry of 46, 10 and 44% of clay, silt and sand, respectively.

11) Line 330: Reviewer comment: explain the rationale for choosing these cultivars for this study.

Original text: Plots consisted of soybean cultivars (NA 5909RG; M 6410IPRO; BRS 5980IPRO; BRS 7180IPRO; BRS 7280RR; BRS 7380RR) and subplots corresponded to four water regimes (WR).

Modified text: Lines 349 to 357 in the new version –

 Plots consisted of soybean cultivars (NA 5909RG; M 6410IPRO; BRS 5980IPRO; BRS 7180IPRO; BRS 7280RR; BRS 7380RR) and subplots corresponded to four water regimes (WR). The cultivars BRS 5980IPRO (roundup ready and intacta technology), BRS 7180IPRO, BRS 7280RR and BRS 7380RR (roundup ready technology) were developed by Embrapa's genetic improvement program, and were selected for this work because of their high yield, production stability, health and wide adaptability to the various grain producing regions of the Brazilian Cerrado. The varieties Nidera cultivar (NA) 5909RG (Glyphosate resistance technology) and Monsoy cultivar (M) 6410IPRO are also recommended for the region and were used in this work as control genotypes.

12) Line 367-373: Reviewer comment: any measurements of soils water content?

Original text: Irrigation at the highest level was carried out as described in the Cerrado Irrigation Monitoring Program [37] to replace evapotranspiration, using agrometeorological indicators of the region and soil type and crop emergence date. The program estimated the reference evapotranspiration based on the equation proposed by Penman-Monteith. The frequency of irrigation was carried out approximately every five days depending on the climatic conditions and the phenological phase of the plants. Two rows of collectors were placed parallel to the irrigation pipe to measure the water applied during each irrigation.

Modified text: Lines 389 to 397 in the new version:

Irrigation at the highest level was carried out as described in the Cerrado Irrigation Monitoring Program [48] to replace evapotranspiration, using agrometeorological indicators of the region and soil type and crop emergence date. The program estimated the reference evapotranspiration based on the equation proposed by Penman-Monteith. The frequency of irrigation was carried out approximately every five days depending on the climatic conditions and the phenological phase of the plants. Two rows of collectors were placed parallel to the irrigation pipe to measure the water applied during each irrigation. Soil water content in the higher and lower water regimes (WR1 and WR4) (Figure 8) just before irrigation and measured at flowering were between 12 and 24%.

Author´s comment: Soil humidity were measured during flowering period using gravimetric measurement.

13) Line 376: Reviewer comment:  maximum quantum yield of photosystem II (Fv/Fm) needs to be measured at predawn when the leaves are fully dark-adapted. Please explain more details on when you did the measurements, and also whether the leaves were dark-adapted or under the light.

Original text: At 70 DAE, net assimilation of CO2 (A), stomatal conductance (gs), transpiration rate (E), and maximum quantum yield of photosystem II (Fv/Fm) were evaluated in the phenological phase R5.1. This evaluation was performed from 8:30 am to 12:30 pm at an irradiance of 1200 μmol photons m-2 s-1 and an external CO2 concentration (Ca) of 400 μmol mol-1 in the air using a portable open-flow gas exchange device (LI -6400xt LI -COR Inc., Lincoln, NE).

Modified text: Lines 404 to 426 in the new version

At 70 DAE, net assimilation of CO2 (A, µmol CO2 m-2 s-1), stomatal conductance (gs, mol H2O m-2 s-1) and transpiration rate (E, mmol H2O m-2 s-1) were evaluated in the phenological phase R5.1. This evaluation was performed from 8:30 am to 12:30 pm at an irradiance of 1200 μmol photons m-2 s-1 and an external CO2 concentration (Ca) of 400 μmol mol-1 in the air using IRGA (infra-red gas analyzer), a portable open-flow gas exchange device (LI -6400xt LI -COR Inc., Lincoln, NE). The chlorophyll fluorescence variable and maximum quantum yield of photosystem II (Fv/Fm), were measured using a modulated portable fluorometer coupled to IRGA. Evaluations were conducted on dark-adapted leaves, for at least 3 h, and the evaluation was performed after 10:30 pm., so that the reaction centers were fully opened (all oxidized primary electron acceptors) with minimum heat loss. Under this condition, it was possible to estimate the initial fluorescence (F0), maximum fluorescence (Fm), and maximum quantum yield of photosystem II [Fv/Fm = (F0-Fm)/Fm)] [49].

Three evaluations were made in each subplot to quantify gas exchange. Evaluations were performed on the youngest fully expanded leaves, with light-adapted. Measurements were made under controlled CO2 concentration, temperature and H2O vapor from the study area, with the reference air being homogenized in a 20L gallon before reaching the leaf chamber. The measurements were performed after the coefficient that combines the variations of carbon dioxide (ΔCO2), water (ΔH2O) and air flow (Δμe) was below 1%. Relative humidity was between 65 and 70%, temperature was between 20 and 25 ºC (night/day), irradiance of 1200 μmol photons m-2 s-1 and external concentration of CO2 of 400 μmol mol-1. The intrinsic water-use efficiency (iWUE) was calculated by the ratio between net assimilation of CO2 and stomatal conductance (A/gs).

14) Reviewer comment:  Gas exchange measurements: please provide detailed environmental conditions including temperature, relative humidity etc.

Modified text : Lines 418 to 425 in the new version

Measurements were made under controlled CO2 concentration, temperature and H2O vapor from the study area, with the reference air being homogenized in a 20L gallon before reaching the leaf chamber. The measurements were performed after the coefficient that combines the variations of carbon dioxide (ΔCO2), water (ΔH2O) and air flow (Δμe) was below 1%. Relative humidity was between 65 and 70%, temperature was between 20 and 25 ºC (night/day), irradiance of 1200 μmol photons m-2 s-1 and external concentration of CO2 of 400 μmol mol-1.

15) Reviewer comment:  Statistical analysis: please also explain the analysis for Figure 1.

Modified text: Lines 102 to107

In the biplot representation of the decomposition into singular values (Figure 1), the main coordinate 1 (latent variable) retained 56% of the multivariate variation in water regimes and cultivars, while coordinate 2 retained 9.9% of the variation. Biplot analysis is a multivariate technique that can be used to analyze interrelationships between a large number of variables and explain these variables in terms of their inherent dimensions and works with one set of variables [27]. According to [28], at least 60% of the total variance must be explained by the first two principal components.

Author´s reply: Any matrix of rank two can be displayed as a biplot which consists of a vector for each row and a vector for each column, chosen so that any element of the matrix is exactly the inner product of the vectors corresponding to its row and to its column. If a matrix is of higher rank, one may display it approximately by a biplot of a matrix of rank two which approximates the original matrix. The biplot provides a useful tool of data analysis and allows the visual appraisal of the structure of large data matrices. It is especially revealing in principal component analysis, where the biplot can show inter-unit distances and indicate clustering of units as well as display variances and correlations of the variables (Gabriel, 1971). Thus, five elements are presented to be taken into account in the future analysis, namely: 1. Points are lines (cases) and vectors are columns. 2. The distance between two cases approximates their similarity. 3. The length of the vector approximates the standard deviation of the variables and the importance of the difference in treatments. 4. The cosine of the angle between two vectors approximates the correlation between the variables. 5. The projection of a case on the axis of a variable approaches the maximum value. Coordinates are the main components. According to Rencher (2002), at least 60% of the total variance must be explained by the first two principal components.

Rencher, A.C. Methods of Multivariate Analysis. 2 th. New York, John Wiley and Sons, 2002.

16) Reviewer comment:  Conclusions. Please write the conclusions as a paragraph. More analyses are needed to figure out which one or combination can predict yield best.

Modfied text: lines 474 to 485 in the new version

Water stress negatively affected gas exchange, vegetative indices, grain yield and protein in different ways in the studied soybean cultivars. Vegetative indices (NDVI and GNDI) are related to the morphophysiological and agronomic responses of soybean and are the most responsive high throughput variables to water stress, which can predict soybean productivity, as demonstrated by biplot and correlation analysis. Soybean cultivars BRS 7280RR and NA 5909RG showed better yields in winter conditions with higher and lower water availability and the soybean cultivar BRS 7280RR showed better tolerance to drought and WUE.

Overall, the study highlights the potential impact of lower water availability on important traits to farmers and consumers and highlight cultivars that are less affected by this condition. Considering the limitations of the work, a study of roots is foreseen in terms of perspectives.

Reviewer 2 Report

Please indicate the method for determination of WUE indicator, as you have measurements/data of some parameters (such as net assimilation of CO2 (A), stomatal conductance (gs), transpiration rate (E), and maximum quantum yield of photosystem II (Fv / Fm) - in section 2.3 WUE), while in section 3.4. Grain yield and quality components - the rows 405, 406 - it is mentioned that  "Water use efficiency (WUE) was calculated using the relationship between grain yield and crop water applied".

For a better understanding of the working method and results, it is advisable to present the phenophases in which certain works were applied or certain determinations were made (row 384 and row 377 - ex phenological phase R5.1).

Please indicate the  N and  humus content of soil both considering their importance for the plant, when establishing the fertilization dose, and for a better understanding of the recorded data and the obtained results.

Author Response

Manuscript: Water stress alters morphophysiological, grain quality and vegetation indices of soybean cultivars.

Reviewer 2:

1) Comment reviwer 2: Please indicate the method for determination of WUE indicator, as you have measurements/data of some parameters (such as net assimilation of CO2 (A), stomatal conductance (gs), transpiration rate (E), and maximum quantum yield of photosystem II (Fv / Fm) - in section 2.3 WUE), while in section 3.4. Grain yield and quality components - the rows 405, 406 - it is mentioned that “Water use efficiency (WUE) was calculated using the relationship between grain yield and crop water applied".

Original text:

Water use efficiency (WUE) was calculated using the relationship between grain yield and crop water applied [34].

Modified text: Lines 449-452.

Water use efficiency (WUE) was calculated using the relationship between grain yield and crop water applied [39]. Additionally, the intrinsic water efficiency (iWUE) was calculated by the relation A/gs, measuring gas exchange using IRGA.

Figure 5. Regression equations and the respective coefficients of determination (R²) of the variables: (a) water use efficiency (WUE), (b)- net assimilation of CO2/stomatal conductance (A/gs or intrinsic water use efficiency iWUE) as a function of water regime (WR1, WR2, WR3 and WR4 - 31, 44, 64, resp. 100% CET replacement) and soybean cultivars (NA 5909RG; M 6410IPRO; BRS 5980IPRO; BRS 7180IPRO; BRS 7280RR; BRS 7380RR).

Lines 245 to 249

Considering the data obtained by iWUE (A/gs, intrinsic water use efficiency), the higher efficiency at leaf level in treatments with lower applied water was obtained (Figure 5b). [41] report that high-yield soybean varieties under drought cope with the shortage of water by enhancing their photoprotective defenses and invest in growth and productivity, linked to a higher intrinsic water use efficiency. [42] indicate photosynthesis and iWUE as traits to be used in genetic improvement strategies.

Author´s reply: Considering your comment as a reviewer, in addition to WUE we included iWUE (A/gs) intrinsic water use efficiency). Results were interesting.

2) Comment reviwer 2: For a better understanding of the working method and results, it is advisable to present the phenophases in which certain works were applied or certain determinations were made (row 384 and row 377 - ex phenological phase R5.1).

Author´s reply: We included physiological analyzes and vegetation indices in R5.1; yield components at harvest.

Adittionally:

Original text:

In both experiments, the same water depth was applied during the first 35 DAE (tillering stage) with an average of 140 mm of water was applied to obtain a homogeneous plant stand. After this period, the "line source" method was applied [42], modified by the introduction of an irrigation bar [43].

Modified text:  

Lines 376 to 379 in the new version

In both experiments, the same water depth was applied during the first 35 DAE (days after emergence), at vegetative stage V3, with an average of 140 mm of water was applied to obtain a homogeneous plant stand. After this period, the "line source" method was applied [46], modified by the introduction of an irrigation bar [47].

3) Comment reviwer 2: Please indicate the N and  humus content of soil both considering their importance for the plant, when establishing the fertilization dose, and for a better understanding of the recorded data and the obtained results.

Original text:

The experiment was carried out between May and September, which coincides with the dry season in the region that allows controlling the water supply to the plants. The soil is classified as Oxisol with clay texture [40], and the soil analysis carried out before conducting the experiment showed the following physicochemical properties at a depth of 0 to 20 cm: pH (CaCl2) of 5.7; 11 mg dm-3 P; 186 mg dm-3 K; 5.77 cmolc dm-3 Ca; 1.83 cmolc dm-3 Mg; 0.02 cmolc dm-3 Al  and granulometry of 46, 10 and 44% of clay, silt and sand, respectively.

Modified text:

Lines 334 to 340 in the new version

The experiment was carried out between May and September, which coincides with the dry season in the region that allows controlling the water supply to the plants. The soil is classified as Oxisol with clay texture [44], and the soil analysis carried out before conducting the experiment showed the following physicochemical properties at a depth of 0 to 20 cm: pH (CaCl2) of 5.7; 11 mg dm-3 P; 186 mg dm-3 K; 5.77 cmolc dm-3 Ca; 1.83 cmolc dm-3 Mg; 0.02 cmolc dm-3 Al; 15,7 mg dm-3 N-NO-3; 2.6% organic matter and granulometry of 46, 10 and 44% of clay, silt and sand, respectively.

Reviewer 3 Report

The study used non-destructive and automated phenotyping methods for selecting drought-tolerant soybean genotypes. They used several morphophysiological, yield, and grain quality traits to select the tolerant cultivars. My major concerns are about the Methods and Results sections. The manuscript is well prepared. I recommend the publication after addressing the below concerns/comments.

Introduction:

The authors focused mainly on the importance of using vegetation indices. However, there should be more about the applications of vegetation indices in measuring drought tolerances. In addition, there should be more justifications for the importance of using the morphophysiological changes as indicators for plant performance, with supportive evidence from similar studies.

Materials and Methods

During the growing seasons (between May and September), some rainfalls were recorded in the two years (2018, 2019). Have the authors accounted for the water received when calculating the different water regimes?

The calculation of the amount of water used in each water regime is not clear. For example, how did the authors calculate WR1 160.1 mm in 2018 and 164.79 mm in 2019?

Have they considered the amount of water evaporated during the irrigation with the sprinkler system? Much water is stuck on the plants evaporated during the irrigation.

The physiological methods are not clear for most of the parameters. For example, how the maximum quantum yield of photosystem II (Fv/Fm) was measured?; how was gas exchange quantified?; how WUE and net assimilation of CO2 was measured? At least, describe in a few sentences each method and refer to relevant references.

Need justification why you selected these drought levels (water deficit). Why do you consider 31, 44, 64, and 100% of crop evapotranspiration replacement were proper levels?

How did you account for water evaporation during sprinkler irrigation?

Have they adjusted for the amount of water evaporated at that time

Results and Discussion

Tables 1 and 2 are taking a big space of the manuscript and make it hard for the reader to extract information. I suggest providing these tables as supplementary Tables, especially since all the information that could be obtained from such tables is well presented in the PCA and correlation table.

Tables 1: Fv/Fm value should be in the range of 0.79 to 0.84 is the approximate optimal value for many plant species, with lowered values indicating plant stress. However, the values reported here are 0.2 – 0.4, which are not realistic. Please check.

The legends of Fig. 1 and 2 are too long with the variable names. I suggest adding the full name of the variables (not the abbreviation) on the diagonal axis; there is enough space to write the full names.

2.5. Protein content and oil content in the grains:  Nothing is mentioned in the Methods section about the quantification of protein and oil contents in the grains.

Fig. 2: Need a high-quality image for the heatmap.

Author Response

Manuscript: Water stress alters morphophysiological, grain quality and vegetation indices of soybean cultivars.

Reviewer 3:

1) General comments:

The study used non-destructive and automated phenotyping methods for selecting drought-tolerant soybean genotypes. They used several morphophysiological, yield, and grain quality traits to select the tolerant cultivars. My major concerns are about the Methods and Results sections. The manuscript is well prepared. I recommend the publication after addressing the below concerns/comments.

2) Introduction:

Reviewer comment: The authors focused mainly on the importance of using vegetation indices. However, there should be more about the applications of vegetation indices in measuring drought tolerances. In addition, there should be more justifications for the importance of using the morphophysiological changes as indicators for plant performance, with supportive evidence from similar studies.

Author´s reply: In addition to papers already in original text, we included more papers (highlighted in yellow) considering drought tolerance, vegetation indices and morphophysiological approaches.

New version of the paper: Lines 72 to 82

The use of vegetation indices can be used to select drought-tolerant soybean cultivars as obtained by [19]. Besides, [20] obtained strong relationships between vegetation indices and plant physiological parameters. Vegetation indices can be used to select drought-tolerant soybean cultivars and other crops at the leaf level and through UAV (non-nano aerial vehicle) platforms [21, 22, 23]. In Addiction, UAV coupled thermal sensors can measure other important features related to canopy temperature showing small differences in leaf temperature associated with water stress [21, 24].

Morphophysiological traits also are indicators of plant performance under water stress, decreasing photosynthesis rate, plant height, number of leaves, pods and shoot dry weight [1, 25]. In addition, grain quality is altered, increasing protein and decreasing oil content [1, 26].

References considering vegetation indices and morphphysiological approaches at the manuscript: new references highlighted in yellow.

CRUSIOL, L. G. T.; NANNI, M. R.; FURLANETTO, R. H.; SIBALDELLI, R. N. R.; CEZAR, E.; SUN, L.; FOLONI, J. S. S.; MERTZ-HENNING, L. M.; NEPOMUCENO, A. L.; NEUMAIER, N.; FARIAS, J. R. B. Classification of Soybean Genotypes Assessed Under Different Water Availability and at Different Phenological Stages Using Leaf-Based Hyperspectral Reflectance. Remote Sensing, v. 13, n. 2, 172, 2021. DOI: 10.3390/rs13020172

LF Pereira, WQ Ribeiro Junior, MLG Ramos, NZ Santos, GF Soares, RACN Casari, O Muller, CJ Tavares, ÉS Martins, U Rascher, CAL Guimarães, AF Pereira, LM Mertz-Henning, CAF Sousa. Physiological changes in soybean cultivated with soil remineralizer in the Cerrado under variable water regimes. Pesquisa Agropecuária Brasileira, 56 (2021), e01455, 10.1590/ S1678-3921.pab2021.v56.01455

Soares, G.F.; Ribeiro Júnior, W.Q.; Pereira, L.F.; Lima, C.A.; Soares, D.D.S.; Muller, O.; Rascher, U.; Ramos, M.L.G. Characterization of wheat genotypes for drought tolerance and water use efficiency. Sci. Agric. 2021, 78, e20190304. https://doi.org/10.1590/1678-992X-2019-0304.

Braga, P.; Crusiol, L.G.T.; Nanni, M.R.; Caranhato, A.L.H.; Fuhrmann, M.B.; Nepomuceno, A.L.; Neumaier, N.; Farias, J.R.B.; Koltun, A.; Gonçalves, L.S.A.; Mertz-Henning, L.M. Vegetation indices and NIR-SWIR spectral bands as a phenotyping tool for water status determination in soybean. Precis. Agric. 2021, 22, 249-266. https://doi.org/10.1007/s11119-020-09740-4.

Zhou, J.; Zhou, J.; Ye, H.; Ali, M.D.L.; Chen, P.C.; Nguyen, H.T.; Chen, P. Yield estimation of soybean breeding lines under drought stress using unmanned aerial vehicle-based imagery and convolutional neural network. Biosyst. Eng. 2021, 204, 90-103. https://doi.org/10.1016/j.biosystemseng.2021.01.017.

Buezo, J.; Sanz-Saez, A.; Morana, J.F.; Sobaa, D.; Aranjuelo, I.; Esteban, R. Drought tolerance response of high‐yielding soybean varieties to mild drought, physiological and photochemical adjustments. Physiol. Plant. 2019, 166, 88-104. https://doi.org/10.1111/ppl.12864.

Wijewardana, C.; Reddy, K.R.; Bellaloui, N. Soybean seed physiology, quality, and chemical composition under soil moisture stress. Food Chem. 2019, 278, 92-100.  https://doi.org/10.1016/j.foodchem.2018.11.035.

Qiu, R.C.; Wei, S.; Zhang, M.; Sun, H.; Li, H.; Liu, G.; Li, M. Sensors for measuring plant phenotyping: A review. Int. J. Agric. & Biol. Eng. 2018, 11, 1-17. https://doi.org/10.25165/j.ijabe.20181102.2696.

Silva, E.E.; Baio, F.H.R.; Teodoro, L.P.R.; Silva Junior, C.A.; Borges, R.S.; Teodoro, P.E. UAV-multispectral and vegetation indices in soybean grain yield prediction based on in situ observation. Remote Sens. Appl. Soc. Environ. 2020, 18, 100318. https://doi.org/10.1016/j.rsase.2020.100318.

Sobejano-Paz, V.; Mikkelsen, T.N.; Baum, A.; Mo, X.; Liu, S.; Köppl, C.J.; Johnson, M.S.; Gulyas, L.; García, M. Hyperspectral and Thermal Sensing of Stomatal Conductance, Transpiration, and Photosynthesis for Soybean and Maize under Drought. Remote Sens. 2020, 12, 3182. https://doi.org/10.3390/rs12193182.

Zhao, Y.; Potgieter, A.B.; Zhang, M.; Wu, B.; Hammer, G.L. Predicting Wheat Yield at the Field Scale by Combining High-Resolution Sentinel-2 Satellite Imagery and Crop Modelling. Remote Sens. 2020, 12, 1024. https://doi.org/10.3390/rs12061024.

Ballester, C.; Zarco-Tejada, P.J.; Nicolás, E.; Alarcón, J.J.; Fereres, E.; Intrigliolo, D.S.; Gonzalez-Dugo, V. Evaluating the performance of xanthophyll, chlorophyll and structure-sensitive spectral indices to detect water stress in five fruit tree species. Precis. Agric. 2018, 19, 178-193. https://doi.org/10.1007/s11119-017-9512-y.

CRUSIOL, L. G. T.; CARVALHO, J. D. F. C.; SIBALDELLI, R. N. R.; NEIVERTH, W.; RIO, A.; FERREIRA, L. C.; PROCÓPIO, S. O.; MERTZ-HENNING, L. M.; NEPOMUCENO, A. L.; NEUMAIER, N.; FARIAS, J. R. B. NDVI variation according to the time of measurement, sampling size, positioning of sensor and water regime in different soybean cultivars. Precision Agriculture, v. 18, p. 470-490, 2017. DOI: 10.1007/s11119-016-9465-6

MAIMAITIYIMING, M.; GHULAM, A.; BOZZOLO, A.; WILKINS, J. L.; KWASNIEWSKI, M. T. Early detection of plant physiological responses to different levels of water stress using reflectance spectroscopy. Remote Sensing, v. 9, n. 7, p. 745, 2017. DOI: 10.3390/rs9070745

Gurumurthy, S.; Sarkar, B.; Vanaja, M.; Lakshmi, J.; Yadav, S. K.; Maheswari, M. (2019). Morpho-physiological and biochemical changes in black gram (Vigna mungo L. Hepper) genotypes under drought stress at flowering stage. Acta Physiol. Plant. 2019, 41, 1-14. https://doi.org/10.1007/s11738-019-2833-x.

Silva, A.N.; Ramos, M.L.G.R.; Ribeiro Júnior, W.Q.R.; Alencar, E.R.; Silva, P.C.; Lima, C.A.; Vinson, C.C.; Silva, M.A.V. Water stress alters physical and chemical quality in grains of common bean, triticale and wheat. Agric. Water Manag. 2020, 231, 106023. https://doi.org/10.1016/j.agwat.2020.106023.

Damm, A., Paul-Limoges, E., Haghighi, E., Simmer, C., Morsdorf, F., Schneider, F.D., Van Der Tol, C., Migliavacca, M., Rascher, U. Remote sensing of plant-water relations, An overview and future perspectives. J. Plant Physiol. 2018, 227, 3-19. https://doi.org/10.1016/j.jplph.2018.04.012.

Basal, O.; Szabó, A.; Veres, S. Physiology of soybean as affected by PEG-induced drought stress. Curr. Plant Biol. 2020, 22, 100135. https://doi.org/10.1016/j.cpb.2020.100135.

Jumrani, K.; Bhatia, V.S. Identification of drought tolerant genotypes using physiological traits in soybean. Physiol. Mol. Biol. Plants. 2019, 25, 697-711. https://doi.org/10.1007/s12298-019-00665-5.

Mwamlima, L.H.; Ouma, J.P.; Cheruiyot, E.K. Physiological response of soybean [Glycine max (L) Merrill] to soil moisture stress. Afr. J. Agric. Res. 2019, 14, 729-739. https://doi.org/10.5897/AJAR2019.13961.

Bertolino, L.T.; Caine, R.S.; Gray, J.E. Impact of stomatal density and morphology on water-use efficiency in a changing world. Front. Plant Sci. 2019, 10, 225. https://doi.org/10.3389/fpls.2019.00225.

Gorthi, A.; Volenec, J.J.; Welp, L.R. Stomatal response in soybean during drought improves leaf-scale and field-scale water use efficiencies. Agric. For. Meteorol. 2019, 276, 107629. https://doi.org/10.1016/j.agrformet.2019.107629.

Buezo, J.; Sanz-Saez, A.; Moran, J.F.; Soba, D.; Aranjuelo, I.; Esteban, R. Drought tolerance response of high-yielding soybean varieties to mild drought: physiological and photochemical adjustments. Physiol Plant. 2019, 166, 88-104. https://doi.org/10.1111/ppl.12864.

Lopez, M.A.; Xavier, A.; Rainey, K. M. Phenotypic Variation and Genetic Architecture for Photosynthesis and Water Use Efficiency in Soybean (Glycine max L. Merr). Front. Plant Sci. 2019, 10. https://doi.org/10.3389/fpls.2019.00680.

Chacon, D.P.; Barajas, E.M.; Esteva, A.G.; Delgado, R.L.; Shibata, J.K.; Valdivia, C.B.P. Biomass remobilization in two common bean (Phaseolus vulgaris) cultivars under water restriction. S. Afr. J. Bot. 2017, 112, 79-88. https://doi.org/10.1016/j.sajb.2017.05.015.

Daughtry, C.S.T.; Walthall, C.L.; Kim, M.S.; Colstoun, E.B.; Mcmurtrey, J.E. Estimating corn leaf chlorophyll concentration from leaf and canopy reflectance. Remote Sens. Environ. 2000, 74, 229-239. https://doi.org/10.1016/S0034-4257(00)00113-9.

Gamon, J.A.; Penuelas, J.; Field, C.B.A. Narrow-waveband spectral index that tracks diurnal changes in photosynthetic efficiency. Remote Sens. Environ. 1992, 41, 35-44. https://doi.org/10.1016/0034-4257(92)90059-S.

Steven, M.D. The sensitivity of the OSAVI vegetation index to Observational Parameters. Remote Sens. Environ. 1998, 63, 49-60. https://doi.org/10.1016/S0034-4257(97)00114-4.

Haboudane, D.; Miller, J.R.; Tremblay, N.; Zarco-Tejada, P.J.; Dextraze, L. Integrated narrowband vegetation indices for prediction of crop chlorophyll content for application to Precis. Agric. Remote Sens. Environ. 2002, 81, 416-426. https://doi.org/10.1016/S0034-4257(02)00018-4.

Materials and Methods

3) Rewiwer comment: During the growing seasons (between May and September), some rainfalls were recorded in the two years (2018, 2019). Have the authors accounted for the water received when calculating the different water regimes?

Author´s Reply: Yes, rainfall was included but in a plant that receives more than 500 mm during the cycle, small rainfall does not bring significant differences.

Modified text lines 362 to 363:

In both years, rainfall was included in the calculations of applied water.

4) Reviewer comment: The calculation of the amount of water used in each water regime is not clear. For example, how did the authors calculate WR1 160.1 mm in 2018 and 164.79 mm in 2019?

Author´s Reply: As it is a field experiment, with annual climatic variations that affect evapotranspiration and small precipitations, there are therefore small differences in irrigation calculations in mm for the same water replacement. The amount of water is calculated and monitored through collectors placed along the plots.

5) Reviewer comment: Have they considered the amount of water evaporated during the irrigation with the sprinkler system? Much water is stuck on the plants evaporated during the irrigation.

Author´s Reply: The water collectors calculate the water applied to the plants and it is not possible to know exactly the exact amount absorbed.

6) Reviewer comment: The physiological methods are not clear for most of the parameters. For example, how the maximum quantum yield of photosystem II (Fv/Fm) was measured?;  how was gas exchange quantified?

Original text: At 70 DAE, net assimilation of CO2 (A), stomatal conductance (gs), transpiration rate (E), and maximum quantum yield of photosystem II (Fv/Fm) were evaluated in the phenological phase R5.1. This evaluation was performed from 8:30 am to 12:30 pm at an irradiance of 1200 μmol photons m-2 s-1 and an external CO2 concentration (Ca) of 400 μmol mol-1 in the air using a portable open-flow gas exchange device (LI -6400xt LI -COR Inc., Lincoln, NE).

Modified text: Lines 394 to 415 in the new version

At 70 DAE, net assimilation of CO2 (A, µmol CO2 m-2 s-1), stomatal conductance (gs, mol H2O m-2 s-1) and transpiration rate (E, mmol H2O m-2 s-1) were evaluated in the phenological phase R5.1. This evaluation was performed from 8:30 am to 12:30 pm at an irradiance of 1200 μmol photons m-2 s-1 and an external CO2 concentration (Ca) of 400 μmol mol-1 in the air using IRGA (infra-red gas analyzer), a portable open-flow gas exchange device (LI -6400xt LI -COR Inc., Lincoln, NE). The chlorophyll fluorescence variable and maximum quantum yield of photosystem II (Fv/Fm), were measured using a modulated portable fluorometer coupled to IRGA. Evaluations were conducted on dark-adapted leaves, for at least 3 h, and the evaluation was performed after 10:30 pm., so that the reaction centers were fully opened (all oxidized primary electron acceptors) with minimum heat loss. Under this condition, it was possible to estimate the initial fluorescence (F0), maximum fluorescence (Fm), and maximum quantum yield of photosystem II [Fv/Fm = (F0-Fm)/Fm)] [49].

Three evaluations were made in each subplot to quantify gas exchange. Evaluations were performed on the youngest fully expanded leaves, with light-adapted. Measurements were made under controlled CO2 concentration, temperature and H2O vapor from the study area, with the reference air being homogenized in a 20L gallon before reaching the leaf chamber. The measurements were performed after the coefficient that combines the variations of carbon dioxide (ΔCO2), water (ΔH2O) and air flow (Δμe) was below 1%. Relative humidity was between 65 and 70%, temperature was between 20 and 25 ºC (night/day), irradiance of 1200 μmol photons m-2 s-1 and external concentration of CO2 of 400 μmol mol-1. The intrinsic water-use efficiency (iWUE) was calculated by the ratio between net assimilation of CO2 and stomatal conductance (A/gs).

8)  how WUE and net assimilation of CO2 was measured? At least, describe in a few sentences each method and refer to relevant references.

Author´s reply: WUE and A/gs (iWUE) were described in the iten 3.4. and assimilation of CO2 was described in the item 3.2 (gas exchange).

Original text: Water use efficiency (WUE) was calculated using the relationship between grain yield and crop water applied [34].

Water use efficiency (WUE) was calculated using the relationship between grain yield and crop water applied [39]. Additionally, the intrinsic water efficiency (iWUE) was calculated by the relation A/gs, measuring gas exchange using IRGA.

Modified text: Lines 449 to 452

9) Reviewer comment Need justification why you selected these drought levels (water deficit). Why do you consider 31, 44, 64, and 100% of crop evapotranspiration replacement were proper levels?

Author´s Reply: The idea of ​​the experiments was to irrigate from several stress level from drastic stress until total replacement of water (please, see fig.8).

10)  Reviewer comment:  How did you account for water evaporation during sprinkler irrigation?

Author´s Reply: We did not make this measurement. Average sprinkler irrigation efficienacy was considered this in the calculation.

11) Reviewer comment : Have they adjusted for the amount of water evaporated at that time

Author´s Reply: In the irrigation calculation the average sprinkler irrigation efficiency is considered.

Results and Discussion

12) Reviewer comment :  Tables 1 and 2 are taking a big space of the manuscript and make it hard for the reader to extract information. I suggest providing these tables as supplementary Tables, especially since all the information that could be obtained from such tables is well presented in the PCA and correlation table.

Author reply: Table 1 and 2 were include in the supplementary material (Tables S1 and S2).

13) Reviewer comment :Tables 1: Fv/Fm value should be in the range of 0.79 to 0.84 is the approximate optimal value for many plant species, with lowered values indicating plant stress. However, the values reported here are 0.2 – 0.4, which are not realistic. Please check.

Author´s reply: Checked and corrected.

Cultivars

WR

Variables

NDVI

SAVI

DVI

GNDVI

NDRE

TCARI

OSAVI

TO

PRI

A

gs

A/gs

E

Fv/Fm

CI

BRS

5980IPRO

31

0.34

0.23

0.14

0.55

0.14

0.16

0.28

0.64

0.30

4.5

0.05

90.7

2.1

0.79

232

44

0.41

0.27

0.17

0.58

0.18

0.17

0.34

0.58

0.32

10.0

0.11

90.9

2.6

0.78

216

64

0.46

0.31

0.19

0.60

0.20

0.19

0.38

0.51

0.32

17.8

0.36

49.4

5.3

0.81

261

100

0.51

0.34

0.20

0.61

0.22

0.21

0.42

0.58

0.33

19.7

0.35

56.3

5.0

0.81

256

BRS

7180IPRO

31

0.42

0.30

0.19

0.58

0.19

0.18

0.35

0.66

0.33

3.7

0.06

61.7

2.6

0.75

241

44

0.49

0.36

0.24

0.61

0.23

0.21

0.43

0.61

0.35

10.9

0.18

60.6

3.5

0.78

226

64

0.57

0.41

0.27

0.62

0.25

0.26

0.49

0.55

0.35

18.3

0.33

55.5

5.3

0.80

249

100

0.76

0.53

0.34

0.69

0.31

0.33

0.64

0.52

0.34

19.9

0.36

55.3

5.1

0.82

254

BRS

7280RR

31

0.39

0.26

0.16

0.58

0.16

0.17

0.32

0.63

0.30

10.0

0.07

142.9

2.3

0.79

184

44

0.44

0.30

0.19

0.60

0.20

0.18

0.37

0.58

0.34

15.3

0.15

102.0

3.2

0.80

204

64

0.50

0.35

0.22

0.61

0.22

0.22

0.43

0.53

0.33

20.7

0.33

62.7

5.2

0.82

248

100

0.56

0.38

0.23

0.62

0.23

0.25

0.47

0.58

0.32

21.2

0.35

60.6

5.2

0.83

251

BRS

7380RR

31

0.37

0.26

0.16

0.55

0.17

0.17

0.31

0.68

0.31

5.3

0.10

53.0

2.4

0.78

188

44

0.44

0.31

0.20

0.58

0.20

0.19

0.38

0.64

0.35

12.7

0.17

74.7

3.5

0.80

217

64

0.48

0.34

0.22

0.59

0.21

0.22

0.41

0.57

0.34

18.8

0.33

57.0

5.1

0.80

257

100

0.64

0.44

0.28

0.61

0.24

0.32

0.54

0.65

0.34

20.7

0.36

57.5

5.1

0.81

250

M

6410IPRO

31

0.39

0.26

0.16

0.56

0.17

0.16

0.33

0.61

0.30

6.9

0.07

98.6

2.2

0.78

283

44

0.45

0.30

0.19

0.60

0.22

0.17

0.37

0.56

0.34

13.9

0.18

77.2

2.9

0.79

216

64

0.49

0.34

0.21

0.61

0.23

0.19

0.41

0.50

0.34

19.0

0.29

65.5

4.6

0.82

256

100

0.55

0.38

0.23

0.63

0.25

0.22

0.46

0.56

0.34

20.9

0.36

58.1

5.2

0.82

251

NA

5909GR

31

0.34

0.22

0.13

0.56

0.15

0.15

0.28

0.60

0.29

7.2

0.06

120.0

3.6

0.80

233

44

0.41

0.27

0.16

0.59

0.19

0.16

0.34

0.56

0.33

14.0

0.15

93.3

3.2

0.82

222

64

0.48

0.32

0.19

0.61

0.21

0.20

0.39

0.54

0.34

20.4

0.31

65.8

4.7

0.83

251

100

0.53

0.35

0.21

0.63

0.23

0.21

0.43

0.55

0.34

21.1

0.36

58.6

5.2

0.84

255

Mean

0.33

0.20

0.60

0.21

0.20

0.40

0.58

0.33

4.0

15.9

0.22

72.2

4.0

0.80

237

SE

0.02

0.01

0.01

0.01

0.01

0.01

0.02

0.01

0.01

0.68

0.03

0.22

0.2

0.01

4.94

CV

2.03

2.13

2.29

0.52

1.84

2.33

2.07

0.84

0.52

1.86

5.67

3.2

3.0

2.19

1.02

14) Reviewer comment: The legends of Fig. 1 and 2 are too long with the variable names. I suggest adding the full name of the variables (not the abbreviation) on the diagonal axis; there is enough space to write the full names.

Author´s reply: We tested and it gets harder to visualize and identify the variables, it gets worse. 

15) Reviewer comment:  2.5. Protein content and oil content in the grains:  Nothing is mentioned in the Methods section about the quantification of protein and oil contents in the grains.

Original text: In addition, grain quality (protein and oil content) and hectoliter weight (HW) were analyzed.

Modified text:

Lines 453 to 462 in the new version

Grain quality (protein and oil content) and hectoliter weight (HW) were analyzed. Protein and oil content (%) were determined in whole grains and without impurities, according to [60]. The analysis were performed in the chemical Analysis Laboratory at EMBRAPA Soybean, by Fourier transform near-infrared spectroscopy (FT-NIR, model Antaris II, ThermoFisher Scientific Waltham, Ma, USA), using 30 g samples of grains, and using an integrating sphere with readings ranging from 1100 to 2500 nm. Mathematical models developed by EMBRAPA Soja in 2011-2012 were used to predict the protein content including 180 standards, correlation coefficient (r) = 0.97, and root mean square error of calibration (RMSEC) = 0.64, and for the oil content: 170 standards, r = 0.98, and RMSEC =0.45.

16) Reviewer commen:t Fig. 2: Need a high-quality image for the heatmap.

Author´s reply: We worked to improve the quality of the figures.

Reviewer 4 Report

Review of plants 1535925 by Cassio Jardim Tavares et al.

The manuscript deals with monitoring different plant traits and relative indices for a set of soybean cultivars in Brazil, as these traits vary with water stress. The authors have the final goal to identify the soybean cultivars that are more resistant to drought.

The manuscript presents some interesting material and analyses. Nevertheless, it can be deeply improved, especially by extending the Section 3 on Materials and Methods. Some analyses, such as those using multivariate statistical analyses (Fig. 1 and Fig. 2) need to better explained, as perhaps not all potential readers are familiar with them. Furthermore, it is unclear how other factors than water may have influenced the findings of the authors’ analyses.  

For these reasons I think that the manuscript should be reassessed after undergoing major revisions.  

Specific comments

Introduction: Perhaps the introduction can mention the aim of the multivariate statistical analysis

LL 77-81 the meaning of this analysis is not clear. Please explain better what you are aiming to.

Fig. 1. The quality of the figure should be improved. Also, there are too many variables in one plot. An effort by the authors should be made to simplify the plot: it is not clear how the information on cultivars (blue circles) should be interpreted when reading the biplot.

Fig. 2. The numbers indication correlation values are difficult to read. This should be improved. Again it is not fully clear what the authors are searching for with this plot.

  1. 337-348 Could have these treatments influenced the results you have obtained? Discuss briefly this aspect

Sect. 3.5 must be extended by explaining the rationale behind the statistical analyses and giving more details on the techniques or at least more references about them. I had the impression that the techniques are applied without having clearly in mind the specific objectives of each analysis.

Fig. 6 improve the quality of this figure

Conclusions should be extended by adding some other results (e.g., a synthesis of the implications of the statistical analysis) and explaining some limitations of the work, as well as some future developments the authors have in mind.

 Minor comments/technical correction  

LL 26-27 this sentence should be rephrased (e.g.: “Rainfall is among the climatic …”)

L35 vegetative-> vegetation

L 76 remove “)”

In the abstract some unexplained abbreviations are present (e.g. BRS 7280RR, …) and WUE.

More references on UAV could be inserted, e.g. :  https://www.sciencedirect.com/science/article/pii/S0034425711003555

https://www.sciencedirect.com/science/article/pii/S0168192313000026

https://ascelibrary.org/doi/10.1061/9780784483466.025

https://www.sciencedirect.com/science/article/pii/S0378377416303201

Author Response

Rewiwer 4

Review of plants 1535925 by Cassio Jardim Tavares et al.

1) General comment: The manuscript deals with monitoring different plant traits and relative indices for a set of soybean cultivars in Brazil, as these traits vary with water stress. The authors have the final goal to identify the soybean cultivars that are more resistant to drought.

2) Reviewer comment: The manuscript presents some interesting material and analyses. Nevertheless, it can be deeply improved, especially by extending the Section 3 on Materials and Methods. Some analyses, such as those using multivariate statistical analyses (Fig. 1 and Fig. 2) need to better explained, as perhaps not all potential readers are familiar with them.

Modified text:

Lines 464 to 473 in the new version

Statistical analysis:

Data were subjected to joint multivariate analysis of variance by harvest. Residuals were tested for multivariate normality using the generalized Shapiro-Wilk test [61] and for homogeneity of covariance matrices using the Box-M test [62]. In two years, treatments (combinations of cultivar and water regime levels) means analyzed graphically in a biplot based on singular value decomposition (SVD) [27] and this allow visualizing the relationship among genotypes and treatments. Pearson correlation analysis (t-test, p<0.05) was per-formed with the residuals. Regression models were fitted to unravel the effect of water regime on the response variables and mean values of the variables were presented in the table (Tables S1 and S2). The statistical analyzes were carried out using the R v3.6.1 software.

3) Reviewer comment: Furthermore, it is unclear how other factors than water may have influenced the findings of the authors’ analyses.  

Author´s reply: The biggest limiting factor of the experiment was water, which leads to an increase in productivity and in the main data as a function of water levels.

4) Reviewer comment: Introduction: Perhaps the introduction can mention the aim of the multivariate statistical analysis

Original text: In the biplot representation of the decomposition into singular values (Figure 1), the main coordinate 1 (latent variable) retained 56% of the multivariate variation in water regimes and cultivars, while coordinate 2 retained 9.9% of the variation.

Modified text:

Lines 104 to 107 in the new version

In the biplot representation of the decomposition into singular values (Figure 1), the main coordinate 1 (latent variable) retained 56% of the multivariate variation in water regimes and cultivars, while coordinate 2 retained 9.9% of the variation. Biplot analysis is a multivariate technique that can be used to analyze interrelationships between a large number of variables and explain these variables in terms of their inherent dimensions and works with one set of variables [27].

5) Reviewer comment:  LL 77-81 the meaning of this analysis is not clear. Please explain better what you are aiming to.

Original text: In the joint multivariate analysis of variance, differences were found for the sources of variation cultivars (p < 0.01), water regime (p < 0.01), year of cultivation (p < 0.01) and for the interaction cultivars x water regime (p < 0.01). There were no differences for cultivars x water regime x cropping year interaction (p = 0.09). The interaction showed that cultivars respond differently to water availability.

Modified text: Lines 99 to 101

In the joint multivariate analysis of variance, differences were found for the sources of variation cultivars (p < 0.01), water regime (p < 0.01), year of cultivation (p < 0.01) and for the interaction cultivars x water regime (p < 0.01). There were no differences for cultivars x water regime x cropping year interaction (p = 0.09). The interaction showed that cultivars respond differently to water availability. Considering the aim of this work this is an opportunity for validate selection methodology and selecting genotypes adapted to growth in hydric stress.

6) Reviewer comment:  Fig. 1. The quality of the figure should be improved. In addition, there are too many variables in one plot. An effort by the authors should be made to simplify the plot: it is not clear how the information on cultivars (blue circles) should be interpreted when reading the biplot.

Author´s reply: We worked in order to improve the quality of the figure. Reduce the number of variables may not be the best option considering that it is an important information.

7) Reviewer comment:  Fig. 2. The numbers indication correlation values are difficult to read. This should be improved. Again it is not fully clear what the authors are searching for with this plot.

Author´s reply: With more variables we can know better the relation among them, in spite of the difficulty in analyze the results. Select genotypes and variables linked with yield/Plant growth would be the main aim.

Modified text: Lines 288 to 295 in the new version

Our results showed that the physiological data A and gs had a high correlation with productivity but could not be done on a large scale, while the vegetative indices NDVI and GNDVI which are high throughput were the ones that showed the highest correlation with yield and can be a useful tool in improvement programs. The vegetation indices had, in general, correlation between them, and one of them could be chosen for the selection process. Among the morphological data NP had a greater correlation with productivity but also a time-consuming evaluation (Fig. 2).

8) Reviewer comment:  337-348. Could have these treatments influenced the results you have obtained? Discuss briefly this aspect.

Original text: The history of the last years of cultivation on the experimental plot is soybean under different water regimes in winter and fallow in summer. The plot was desiccated 20 days before sowing with glyphosate at a dose of 1440 g.e.a ha-1. Soybean seeds were previously inoculated with Bradyrhizobium japonicum (strain SEMIA 5080) at a dose of 100 ml per 50 kg of seed. The seeds were sown mechanically on 2 June 2018 and 23 May 2019 under no-tillage systems, with 16 seeds per meter. For basic fertilization, 300 kg ha-1 fertilizer with formula 04-30-16 (N, P2O5, K2O) was used. Phytosanitary treatments were performed for the cucurbit beetle (Diabrotica speciosa) control; the insecticide thiamethoxam + lambda-cyhalothrin was applied at a dosage of 14.1 g + 10.6 g ha-1 on the 10th and 20th day after soybean emergence (DAE) in 2018 and on the 12th and 24th DAE in 2019. In addition, glyphosate was applied at a dose of 720 g.e.a. ha-1 for weed control at 18 DAE in both years.

Author´s reply:  This small difference in days in the plantings of different years would not be a problem considering that we planted at the same planting time (dry winter). These differences in days between years in insect or weed control depend on the need of pesticides control that occurs each year. The important is that enough fertilization, disease, weeds or insects’ control cannot reduce yield and in this way, water is the main limiting factor.

9) Reviewer comment:  Sect. 3.5 must be extended by explaining the rationale behind the statistical analyses and giving more details on the techniques or at least more references about them. I had the impression that the techniques are applied without having clearly in mind the specific objectives of each analysis.

Modified text:

Lines 464 to 473 in the new version

Data were subjected to joint multivariate analysis of variance by harvest. Residuals were tested for multivariate normality using the generalized Shapiro-Wilk test [61] and for homogeneity of covariance matrices using the Box-M test [62]. In two years, treatments (combinations of cultivar and water regime levels), and means were analyzed graphically in a biplot based on singular value decomposition [27] and this allow visualizing the relationship among genotypes and treatments. Pearson correlation analysis (t-test, p<0.05) was per-formed with the residuals. Regression models were fitted to unravel the effect of water regime on the response variables and mean values of the variables were presented in the table (Tables S1 and S2). The statistical analyzes were carried out using the R v3.6.1 software.

10) Reviewer comment:  Fig. 6 improve the quality of this figure

Author´s reply: We have been working to improve the quality of the figures.

11) Reviewer comment:  Conclusions should be extended by adding some other results (e.g., a synthesis of the implications of the statistical analysis) and explaining some limitations of the work, as well as some future developments the authors have in mind.

Modified text:

Lines 475 to 486 in the new version

Water stress negatively affected gas exchange, vegetative indices, grain yield and protein in different ways in the studied soybean cultivars. Vegetative indices (NDVI and GNDI) are related to the morphophysiological and agronomic responses of soybean and are the most responsive high throughput variables to water stress, which can predict soybean productivity, as demonstrated by biplot and correlation analysis. Soybean cultivars BRS 7280RR and NA 5909RG showed better yields in winter conditions with higher and lower water availability and the soybean cultivar BRS 7280RR showed better tolerance to drought and WUE.

Overall, the study highlights the potential impact of lower water availability on important traits to farmers and consumers and highlight cultivars that are less affected by this condition. Considering the limitations of the work, a study of roots is foreseen in terms of perspectives.

 Minor comments/technical correction  

12) Reviewer comment:  LL 26-27 this sentence should be rephrased (e.g.: “Rainfall is among the climatic …”)

Original text: Among the climatic factors that most affect production is rainfall, as occurs in

the Brazilian Cerrado.

Modified text: Rainfal is among the climatic factors that most affect production is rainfall, as occurs in the Brazilian Cerrado.

13) Reviewer comment:  L35 vegetative-> vegetation

Original text: Water stress had different effects on gas exchange, vegetative indices, grain yield and chemical composition among the cultivars.

Modified text: Water stress had different effects on gas exchange, vegetation indices, grain yield and chemical composition among the cultivars.

14) Reviewer comment:   L 76 remove “)”

Original text: The mean values of vegetation indices, morphophysiological evaluations, grain quality and yield in 2018 and 2019 are presented in Tables 1 and 2).

Modified text: The mean values of vegetation indices, morphophysiological evaluations, grain quality and yield in 2018 and 2019 are presented in Tables S1 and S2.

15) Reviewer comment:   In the abstract some unexplained abbreviations are present (e.g. BRS 7280RR, …) and WUE.

Original text: Abstract: Among the climatic factors that most affect production is rainfall, as occurs in the Brazilian Cerrado. Non-destructive and automated phenotyping methods are fast and efficient for genotype selection. The objective of this work was to evaluate, under field conditions, the morphophysiological changes, yield, and grain quality of soybean under water stress in the Brazilian Cerrado. The plots were composed of six soybean cultivars and the subplots of four water regimes, corresponding to 31, 44, 64 and 100% of crop evapotranspiration replacement. The experiments were conducted from May to September 2018 and 2019. An irrigation system with a bar of sprinklers with different flow rates was used. Gas exchange, vegetation indices measured through a hyperspectral sensor embedded in a drone, yield and grain quality were evaluated. Water stress had different effects on gas exchange, vegetative indices, grain yield and chemical composition among the cultivars. BRS 7280RR and NA 5909RG are yield stable and have greater tolerance to drought. BRS 7280RR showed higher tolerance to drought and WUE. Vegetative indices correlated to the morphophysiological traits and were the most responsive variables to water stress, which can predict soybean yield and can be used as a tool in selection for this purpose.

Modified text: Abstract: Rainfal is among the climatic factors that most affect production is rainfall, as occurs in the Brazilian Cerrado. Non-destructive and automated phenotyping methods are fast and efficient for genotype selection. The objective of this work was to evaluate, under field conditions, the morphophysiological changes, yield, and grain quality of soybean under water stress in the Brazilian Cerrado. The plots were composed of six soybean cultivars and the subplots of four water regimes, corresponding to 31, 44, 64 and 100% of crop evapotranspiration replacement. The experiments were conducted from May to September 2018 and 2019. An irrigation system with a bar of sprinklers with different flow rates was used. Gas exchange, vegetation indices measured through a hyperspectral sensor embedded in a drone, yield and grain quality were evaluated. Water stress had different effects on gas exchange, vegetative indices, grain yield and chemical composition among the cultivars. Embrapa cultivar (BRS) 7280 Roundup ready (RR) and Nidera cultivar (NA) 5909 Glifosate resistant (RG) are yield stable and have greater tolerance to drought. BRS 7280RR showed higher tolerance to drought and water use efficiency (WUE). Vegetative indices correlated to the morphophysiological traits and were the most responsive variables to water stress, which can predict soybean yield and can be used as a tool in selection for this purpose.

16) Review comment More references on UAV could be inserted, e.g. 16.1.:  https://www.sciencedirect.com/science/article/pii/S0034425711003555

16.2. https://www.sciencedirect.com/science/article/pii/S0168192313000026

16.3. https://ascelibrary.org/doi/10.1061/9780784483466.025

16.4. https://www.sciencedirect.com/science/article/pii/S0378377416303201

Authors reply. References included in the manuscript.

Authros coment: we included all these references.

Round 2

Reviewer 1 Report

Tavares et al. Water stress alters morphophysiological, grain quality and vegetation indices of soybean cultivars

The authors have addressed most of my concerns, and I only have these following minor comments. I also would suggest the format and language be checked carefully by a native English speaker.

Line 36-37: greater tolerance to drought compared to all other cultivars? Comparative needs to mention which you are comparing to.

Line 85-90: please put your objective before your hypothesis.

Line 85-94: Usually, a paragraph needs more than one sentence. Combine those paragraphs with only one sentence with other paragraphs.

Line 396-396: Please clarify the difference in soil water potential between WR1 and WR4. Is WR1 12% and WR4 24%?

Line 479-481: better tolerance to 481 drought and WUE “compared to other cultivars”.

Conclusions. One or two sentences explaining whether the Cerrado is predicted to be drier can show the future use of the information here, and the importance of this study.

Author Response

Reviewer 1

1) The authors have addressed most of my concerns, and I only have these following minor comments. I also would suggest the format and language be checked carefully by a native English speaker.

Author Reply:

A native english speaker reviewed the manuscript.

2) Line 36-37: greater tolerance to drought compared to all other cultivars? Comparative needs to mention which you are comparing to.

Author Reply:

We state in the text that the comparison occurred with the other cultivars studied in this manuscript (Line 38).

3) Line 85-90: please put your objective before your hypothesis.

Author reply: Done.

4) Line 85-94: Usually, a paragraph needs more than one sentence. Combine those paragraphs with only one sentence with other paragraphs.

Author reply: We combined two paragraphs. Thus, the objective of this work was to validate high throughput sensors and morphophysiological measurements as tools for the selection of soybean cultivars for yield and grain quality under water stress. This work hypothesizes that reducing water availability reduces grain yield and quality and non-destructive variables are important for soybean genotype selection under water stress.

5) Line 396-396: Please clarify the difference in soil water potential between WR1 and WR4. Is WR1 12% and WR4 24%?

Author reply: We calculated the soil water potential (Soil water content in the higher and lower water regimes (WR1 and WR4) (Figure 8) just before irrigation and measured at flowering were between 15 (-1500 Kpa) and 24% (-50 Kpa). Line 403/405 in the new version of the manuscript.

6) Line 479-481: better tolerance to drought and WUE “compared to other cultivars”.

Author reply:

The BRS7280RR genotypes showed better drought tolerance and water use efficiency than the other genotypes, being able to be used in both humidity conditions. Line 485/487 in the new version.

7) Conclusions. One or two sentences explaining whether the Cerrado is predicted to be drier can show the future use of the information here, and the importance of this study.

Author reply:

With the advent of climate change affecting the Cerrado, this result could be useful in a likely increase in drought events in the region. Overall, the study highlights the potential impact of lower water availability on important traits to farmers and consumers and highlights cultivars less affected by this condition. Lines 487/490 in the new version of the manuscript.

Reviewer 3 Report

The authors addressed all my concerns. I am very satisfied with the modified version

Thank you

Author Response

Thanks.

Reviewer 4 Report

The authors have improved their manuscript, following my suggestions. 

There are a few technical corrections to apply (see attached PDF). 

The authors should read again carefully the manuscript and try to further improve clarity about how the results of the statistical analyses can be interpreted. 

Author Response

Reviewer 4:

Reviewer comment: The authors should read again carefully the manuscript and try to further improve clarity about how the results of the statistical analyses can be interpreted. 

Author reply: We read all the manuscript carefully and we improved results and discussion.

PDF corrections:

Attached PDF corrections:

1) Reviewer comment: Abstract: It is correct to say tolerance to WUE?

Author reply: In the new version we included: BRS 7280RR showed higher tolerance to drought and higher water use efficiency (WUE) than all other tested cultivars. (line 37/38 of the new version of the manuscript).

Author comment: Drought tolerance is the capaciaty to growth and produce under hydric stress. WUE is the graqin productiaon for mm of water applied. We also included physiological WUE (iWUE) that is the relation between photossinthesis and stomatic conductance (A/Gs).

2) Reviewer comment: Glycine max (unclear key word).

Author reply: We removed Glycine max from the keywords.(line 42 of the new version).

3) Reviewer correction: Considering yield under drought Embrapa (reviewer asked to exclude the part in yellow)  cultivar (BRS) 7280 Roundup ready (RR) and  Nidera cultivar (NA) 5909 (Glyphosate resistance) RG are yield stable and have greater tolerance to 37 drought. BRS 7280RR showed higher tolerance to drought and water use efficiency (WUE). The reviewer requested that we delete the part in yellow (line 36).

Author comment: Done. Embrapa cultivar BRS 7280 Roundup ready (RR) and Nidera cultivar NA 5909 RG (Glyphosate Resistance) are yield stable and have greater tolerance to drought. (line 36;37 of the new version).

4) Reviewer correction: Reviewer asked to exclude “ Figure 1” from the text in the line 125.

Author reply: Done. Variables related to vegetation indices, gas exchange and WUE are important in distinguishing between treatments (figure 1 excluded from here) , as they have higher weights (length of arrows). This indicates that these variables are most indicative of crop performance under water stress [23]. On the other hand, stem diameter, oil content, hectoliter weight, germination and grains per pod showed low weight in the distinction between water regimes and cultivars (Figure 1).

5) Reviewer correction: Reviewer asked to replace vegetative for vegetation index.

Author reply: Done.  We changed vegetative for vegetation index in all the manuscript.

5) Reviewer comment: The authors should read again carefully the manuscript and try to further improve clarity about how the results of the statistical analyses can be interpreted. 

Author reply: We read all the anuscript carefully and we improved results and discussion.